# Linking rattiness, geography and environmental degradation to spillover *Leptospira* infections in marginalised urban settings: An eco-epidemiological community-based cohort study in Brazil

Max T Eyre[1,2]*[‡], Fábio N Souza[3], Ticiana SA Carvalho-Pereira[3], Nivison Nery[3], Daiana de Oliveira[3], Jaqueline S Cruz[3], Gielson A Sacramento[3], Hussein Khalil[3,4], Elsio A Wunder[5,6], Kathryn P Hacker[7], José E Hagan[8], James E Childs[5,6], Mitermayer G Reis[3,5], Mike Begon[9], Peter J Diggle[1], Albert I Ko[5,6], Emanuele Giorgi[1†], Federico Costa[1,3,5,6†]

[1]Centre for Health Informatics, Computing, and Statistics, Lancaster University Medical School, Lancaster, United Kingdom; [2]Liverpool School of Tropical Medicine, Liverpool, United Kingdom; [3]Institute of Collective Health, Federal University of Bahia, Salvador, Brazil; [4]Swedish University of Agricultural Sciences, Umeå, Sweden; [5]Oswaldo Cruz Foundation, Brazilian Ministry of Health, Salvador, Brazil; [6]Department of Epidemiology of Microbial Diseases, Yale School of Public Health, New Haven, United States; [7]University of Pennsylvania, Philadelphia, United States; [8]World Health Organization (WHO) Regional Office for Europe, Copenhagen, Denmark; [9]Department of Evolution, Ecology and Behaviour, University of Liverpool, Liverpool, United Kingdom

*For correspondence:
maxeyre3@gmail.com

†These authors contributed equally to this work

Present address: ‡Centre for Health Informatics, Computing and Statistics, Lancaster Medical School, Lancaster University, Lancaster, United States

## Abstract

**Background:** Zoonotic spillover from animal reservoirs is responsible for a significant global public health burden, but the processes that promote spillover events are poorly understood in complex urban settings. Endemic transmission of *Leptospira*, the agent of leptospirosis, in marginalised urban communities occurs through human exposure to an environment contaminated by bacteria shed in the urine of the rat reservoir. However, it is unclear to what extent transmission is driven by variation in the distribution of rats or by the dispersal of bacteria in rainwater runoff and overflow from open sewer systems.

**Methods:** We conducted an eco-epidemiological study in a high-risk community in Salvador, Brazil, by prospectively following a cohort of 1401 residents to ascertain serological evidence for leptospiral infections. A concurrent rat ecology study was used to collect information on the fine-scale spatial distribution of 'rattiness', our proxy for rat abundance and exposure of interest. We developed and applied a novel geostatistical framework for joint spatial modelling of multiple indices of disease reservoir abundance and human infection risk.

**Results:** The estimated infection rate was 51.4 (95%CI 40.4, 64.2) infections per 1000 follow-up events. Infection risk increased with age until 30 years of age and was associated with male gender. Rattiness was positively associated with infection risk for residents across the entire study area, but this effect was stronger in higher elevation areas (OR 3.27 95% CI 1.68, 19.07) than in lower elevation areas (OR 1.14 95% CI 1.05, 1.53).

**Conclusions:** These findings suggest that, while frequent flooding events may disperse bacteria in regions of low elevation, environmental risk in higher elevation areas is more localised and directly driven by the distribution of local rat populations. The modelling framework developed may have broad applications in delineating complex animal-environment-human interactions during zoonotic spillover and identifying opportunities for public health intervention.

**Funding:** This work was supported by the Oswaldo Cruz Foundation and Secretariat of Health Surveillance, Brazilian Ministry of Health, the National Institutes of Health of the United States (grant numbers F31 AI114245, R01 AI052473, U01 AI088752, R01 TW009504 and R25 TW009338); the Wellcome Trust (102330/Z/13/Z), and by the Fundação de Amparo à Pesquisa do Estado da Bahia (FAPESB/JCB0020/2016). MTE was supported by a Medical Research UK doctorate studentship. FBS participated in this study under a FAPESB doctorate scholarship.

## Editor's evaluation

In their work, the authors present a novel geostatistical framework allowing for modelling complex animal-environment-human interactions during zoonotic spillover. The presented case relates to zoonotic spillover of Leptospira infections in a marginalised urban setting in Salvador, Brazil. The outcomes of such applications could contribute to inform public health interventions. The methodological approach is to be applauded and can be of benefit beyond the study of zoonotic spillover.

## Introduction

Zoonotic spillover, the transmission of pathogens from infected vertebrate animals to humans, is responsible for a significant public health burden globally. Understanding the processes that promote spillover transmission is essential for improving our ability to predict and prevent spillover events, but for many zoonoses, such as *Leptospira interrogans*, *Escherichia coli* O157 and *Giardia* spp., they are poorly understood (*Plowright et al., 2017*). This is due to the complex nature of the spillover system, in which the probability of transmission is governed by dynamic interactions in space and time between ecological, epidemiological, behavioural, and immunological factors that determine pathogen pressure, exposure and host susceptibility. Zoonotic spillover research must explore interactions between the environment, disease reservoirs and local epidemiology, presenting two central challenges: (i) the need for transdisciplinary studies at the animal-human disease interface (a One Health approach) that accurately collect data on multiple components of the spillover process at common temporal and spatial scales at which these events take place; (ii) the development of integrative approaches to jointly analyse these diverse datasets within a spatially and temporally explicit framework (*Plowright et al., 2017*; *Becker et al., 2019*; *Dhewantara et al., 2019*).

Leptospirosis, a neglected zoonotic disease caused by pathogenic bacteria from the genus *Leptospira*, is an important example of zoonotic spillover. Globally, it is estimated to cause more than one million cases and over 58,000 deaths each year (*Costa et al., 2015a*), with an annual global burden of 2.9 million disability-adjusted life years (DALYs) (*Torgerson et al., 2015*). This burden falls heavily on marginalised urban populations in low- and middle-income countries who live in areas characterised by high population density, poor quality housing and inadequate provision of healthcare, sanitation, and waste management services. In these settings, leptospiral infection occurs through contact with water or soil contaminated with leptospires shed in the urine of the principal reservoir, the Norway rat (*Rattus norvegicus*; *Bierque et al., 2020*). These areas produce the socio-ecological conditions that allow rodent populations to proliferate and leptospires to persist for long periods in the environment (*Goarant, 2016*). Residents consequently have frequent, intense and largely unavoidable exposure to the contaminated environment, often exacerbated by their geographical vulnerability to flooding events (*Lau et al., 2010*). In response, the World Health Organisation (WHO) has convened the Leptospirosis Burden Epidemiology Reference Group (LERG) which has recommended 'Targeted intervention based on the improved knowledge of disease ecology' (*WHO, 2010*), highlighting the current knowledge gap for *Leptospira* transmission mechanisms and target points for effective intervention.

Multiple studies have helped to elucidate key aspects of the *Leptospira* transmission cycle in urban settings, identifying socioeconomic vulnerability, household environment and behavioural exposures as important determinants of infection risk (*Reis et al., 2008*; *Felzemburgh et al., 2014*; *Hagan et al.,*

*2016*; *Khalil et al., 2021*; *Barcellos and Sabroza, 2000*; *Barcellos and Sabroza, 2001*; *Mwachui et al., 2015*; *Keenan et al., 2010*; *Goarant, 2016*; *Prabhakaran et al., 2014*; *Briskin et al., 2019*). However, these variables have been unable to explain fine-scale spatial variation in risk (*Reis et al., 2008*; *Hagan et al., 2016*). This is likely to be driven by the high spatial and temporal heterogeneity in environmental risk, observed in recent studies of *Leptospira* in soil, and surface and sewage waters (*Schneider et al., 2018*; *Casanovas-Massana et al., 2018*; *Bierque et al., 2020*). These findings lead to two key questions: (i) to what extent does environmental contamination by localised rat shedding drive infection risk, rather than exposure to leptospires that have been dispersed by rainwater runoff and overflowing sewer systems; and (ii) how does this change across the geography of a community, for example at different elevation levels?

Establishing a dynamic link between rats, the environment and *Leptospira* transmission is complicated by the difficulty of measuring and modelling the rat contamination process. However, urban Norway rats have been found to have high *Leptospira* prevalence and shedding rates worldwide (*Pellizzaro et al., 2019*; *Costa et al., 2014a*; *Boey et al., 2019*; *Yusof et al., 2019*; *Krøjgaard et al., 2009*; *Costa et al., 2015b*; *de Faria et al., 2008*). This suggests that rat abundance may be predictive of environmental risk, and could be used as a proxy for this shedding process. While several studies have identified associations between infection risk and household rat sightings and infestation (*Reis et al., 2008*; *Costa et al., 2014b*; *Hagan et al., 2016*; *Costa et al., 2021*; *Pellizzaro et al., 2019*; *Bhardwaj et al., 2008*), their ability to explore fine-scale spatial variation in risk was limited by a reliance on household infestation surveys or aggregation of incidence and abundance indices to a common coarse spatial scale. All modelled abundance as a regression covariate, thereby not accounting for uncertainty in its measurement. The absence of methods applied to formally integrate abundance and spillover infection data is an issue for rodent-borne zoonoses more widely (*Bordes et al., 2015*; *Dhewantara et al., 2019*).

There is no gold-standard index of abundance and field teams use a range of imperfect indices, such as traps, infestation surveys and track plates. In our previous work, we developed a multivariate generalized linear geostatistical model for joint spatial modelling of multiple imperfect abundance indices (*Eyre et al., 2020*). We use the term 'abundance' here to denote all ecological processes that are associated with animal abundance and measured by abundance indices, for example animal presence, density and activity, and that may be useful to quantify exposure to a zoonotic disease of interest. This methodology was then used to model the spatial distribution of 'rattiness', our proxy for rat abundance, at a fine scale within a community in Salvador, Brazil (*Eyre et al., 2020*). The spatial distribution of rattiness was highly heterogeneous, suggesting that it could be a driver of micro-heterogeneity in infection risk.

To analyse reservoir host abundance (as defined previously) and infection data at fine spatial scales, we propose that a framework should (i) account for spatial correlation in human and reservoir host data; (ii) jointly model multiple imperfect indices of abundance while accounting for the appropriate sampling distribution of each index; (iii) account for uncertainty in abundance indices, (iv) allow for the prediction of abundance and infection risk at all locations within the study area, and (v) quantify the uncertainty associated with those predictions. Several studies have attempted to model spatial associations between disease reservoir or vector abundance and human infection for leptospirosis (*Hurd et al., 2017*; *Lau et al., 2016*; *Mayfield et al., 2018*), tularemia (*Rotejanaprasert et al., 2018*) Lyme disease (*Nicholson and Mather, 1996*) West Nile Virus (*Winters et al., 2008*) dengue fever (*Cromwell et al., 2017*) and Lassa fever (*Fichet-Calvet et al., 2007*). However, none of the approaches used satisfy all five of the above conditions. The development of new tools for the joint spatial analysis of abundance and human infection may consequently be beneficial for the study of other zoonoses and vector-borne diseases (*Eisen and Eisen, 2008*).

The aim of this study was to develop a flexible modelling framework for zoonotic spillover to explore whether rattiness, acting as a proxy for local leptospiral contamination by Norway rats, can explain spatial heterogeneity in leptospiral transmission in a high-risk urban community in Brazil where 80% of rats are estimated to be actively shedding the bacteria (*Costa et al., 2015b*; *de Faria et al., 2008*). We extend the rattiness framework of *Eyre et al., 2020* to include human infection risk. We describe findings from a transdisciplinary eco-epidemiological study which comprises a prospective community-based cohort study with two serosurveys and a fine-scale rat ecology study. The ecology study was used to collect information on the spatial distribution of rat abundance, our exposure of

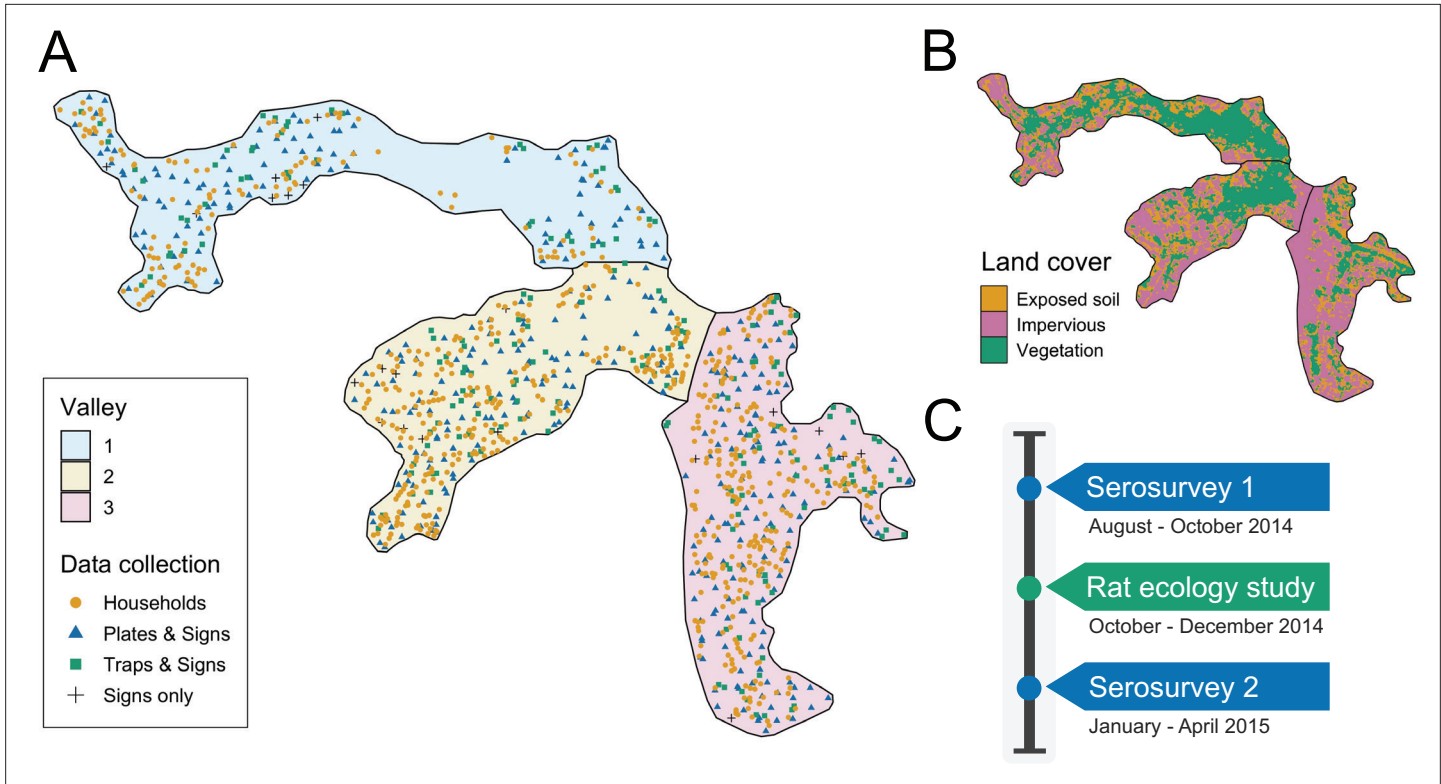

**Figure 1.** Study site and timeline. (**A**) Map of the three valleys within the study site in Pau da Lima, with household locations for the serosurveys marked as orange circles. Locations sampled in the the rat ecology study are shown for each of the rat abundance indices as follows: Plates & Signs (track plates, burrows, faeces and trails), Traps & Signs (traps, burrows, faeces and trails) and Signs only (burrows, faeces, and trails); (**B**) Land cover classification map (impervious cover is defined as man-made structures e.g. pavement and buildings); (**C**) Study timeline for the two community serosurveys and rat ecology study.

interest, in the period between the two surveys using multiple abundance indices. Then, we explore associations between infection risk, rattiness and a range of measured environmental and individual risk factors.

## Materials and methods
### Study design
#### Study area
The study was conducted in Pau da Lima community (13°32′53.47″ S; 38°43′51.10″ W), a marginalised informal settlement located in the city of Salvador, Northeast Brazil. The study site has an area of $0.25 km^2$ and is characterised by three connected valleys with large elevation gradients, high population density and a heterogeneous environment of vegetation, paved surfaces and exposed soil (*Figure 1*). There are significant gradients in socioeconomic status and infrastructure quality over small elevation increases - with the most marginalised members of the community living at lower elevations. The community suffers from low quality housing, poor provision of waste management services and inadequate drainage and sanitation systems (*Hagan et al., 2016*; *Hacker et al., 2020*). Residents are consequently often unable to avoid intense exposure with mud and floodwater. These factors result in abundant rat populations (*Eyre et al., 2020*) and a high estimated annual *Leptospira* infection rate of 35.4 (95% CI, 30.7, 40.6) infections per 1000 annual follow-up events (*Hagan et al., 2016*). For this reason, Pau da Lima has become an exemplar for investigating urban *Leptospira* transmission in Brazil over the last 15 years.

#### Serosurveys
We conducted a prospective community cohort study with two serosurveys carried out in August-October 2014 and January-April 2015. After an initial census of the study site, all ground floor

households were visited and inhabitants who met the eligibility criteria of ≥5 years of age who had slept ≥3 nights in the previous week in a study household were invited to join the study. This study focussed on ground floor households because they are vulnerable to flooding and consequently at high risk for leptospiral transmission. The criterion for determining whether a resident is currently living at a household location is commonly applied in this context to account for resident mobility.

During each survey trained phlebotomists collected blood samples from participants and administered a modified version of the standardised questionnaire used previously (*Costa et al., 2014b*; *Hagan et al., 2016*). Information was collected on demographic and socioeconomic indicators, household environmental characteristics and exposures to potential sources of environmental contamination in the previous six months (the average time between the two serosurveys). Study data were collected and managed using REDCap electronic data capture tools (*Harris et al., 2009*) and all individual data were anonymised. The locations of sampled households are shown in *Figure 1* - panel A. If an individual was not found during a sample collection visit their house was revisited at least five times on different days of the week.

The microscopic agglutination test (MAT) was used to determine titers of agglutinating antibodies against pathogenic *Leptospira* in sera obtained from the blood samples collected in each serosurvey. Serological samples were reacted with a panel of two *Leptospira* reference strains that are dominant in Pau da Lima: *Leptospira interrogans* serovars Copenhageni (COPL1) and Cynopteri 3522 C (C3522C). These two strains have been shown to have the same performance in identifying MAT seroconversion in our prospective studies as the WHO recommended battery of 19 reference serovars. When agglutination was observed at a dilution of 1:50, the sample was titrated in serial twofold dilutions to determine the highest agglutination titer. The study outcome of leptospiral infection was defined as seroconversion, an MAT titer increase from negative to ≥1:50, or a fourfold increase in titer for either serovar between paired samples from cohort subjects. All laboratory analyses were performed in the Laboratory Pathology and Molecular Biology at Fiocruz, Salvador. As part of quality control procedures two independent evaluations were conducted by Yale University for all infected subjects and 8% of all samples, with high concordance between results.

## Rat ecology study

To estimate exposure risk due to local rat contamination between the two serosurveys, a cross-sectional rat ecology study was conducted from October to December 2014. As has been described previously (*Eyre et al., 2020*), the aim of this study was to collect data on the fine-scale spatial variation in rat reservoir population abundance. Data were collected for five indices of rat abundance: live trapping, track plates, number of active burrows present, presence of faecal droppings and presence of trails. Rat trapping was carried out at 189 locations, randomly distributed across the study area (see *Panti-May et al., 2016*). Two traps were deployed for 4 consecutive 24 hr trapping periods at each location. Trapping success and trap closure without a rat, a common malfunction, were recorded after each 24 hr period. Track plates were placed at 415 locations for two consecutive 24 hr periods following the standardised protocol for placement and survey developed and validated previously (*Hacker et al., 2016*), with five plates placed at each location in the shape of a 'five' on a die. After each 24 hr period, plates were repainted and any lost plates were recorded and replaced. On the first day of trapping or plate placement, a survey for signs of rat infestation, adapted from the *Centers for Disease Control and Prevention, 2006* and validated in the study area (*Costa et al., 2014b*), was conducted within an area of 10 m radius around each trapping or plate location to record the number of active burrows and the presence of faecal droppings and trails. In total, 595 independent locations were sampled for traps, track plates and the three survey indices for signs of rat infestation. The spatial distribution of these locations is shown in *Figure 1* - panel A. At 21 locations, theft and local gang violence meant that data for track plates and traps was not collected and only the three survey indices for signs of rat infestation were used.

## Environmental data

In addition to the environmental survey conducted at each household location, we also collected information for three spatially continuous environmental variables: elevation relative to the bottom of each valley, distance to large public refuse piles and the proportion of land cover classified as impervious (man-made structures) within a 30 m radius. The land cover variable was created from Digital

Globe's WorldView-2 satellite imagery (8 bands) taken on February 17, 2013 which was classified using a maximum likelihood supervised algorithm and validated with ground truthed data collected from 20 randomly selected sites of size 5 m by 5 m. The classification map is shown in *Figure 1* - panel B.

## Ethics

Participants were enrolled according to written informed consent procedures approved by the Institutional Review Boards of the Oswaldo Cruz Foundation and Brazilian National Commission for Ethics in Research, Brazilian Ministry of Health (CAAE: 01877912.8.0000.0040) and Yale University School of Public Health (HIC 1006006956).

For the rat ecology study, the ethics committee for the use of animals from the Oswaldo Cruz Foundation, Salvador, Brazil, approved the protocols used (protocol number 003/2012), which adhered to the guidelines of the American Society of Mammalogists for the use of wild mammals in research (*Sikes and Gannon, 2011*) and the guidelines of the American Veterinary Medical Association for the euthanasia of animals (*Leary et al., 2013*). These protocols were also approved by the Yale University's Institutional Animal Care and Use Committee (IACUC), New Haven, Connecticut (protocol number 2012–11498).

## Joint modelling rat abundance and human infection: the rattiness-infection framework

The developed geostatistical modelling framework jointly models multiple rat abundance indices as measurements of a common latent process, called rattiness. Rattiness at each household location contributes to the risk of infection for all inhabitants, in addition to other measured individual or household-level explanatory variables.

We model the rat abundance data following a similar structure to that previously outlined (*Eyre et al., 2020*). Let $R(x)$ denote a spatially continuous stochastic process, representing rattiness. The rat data then consist of a set of outcomes $Y_i = (Y_{i,k} : k = 1, \ldots, 5)$, for $i = 1, \ldots, N_r$, collected at a discrete set of locations $X = \{x_i : i = 1, ..., N_r\}$. The outcome variables $Y_k : k = 1, ..., 5$ are the set of five rat abundance indices that provide information about $R(x)$: traps ($k = 1$), track plates ($k = 2$), number of burrows ($k = 3$), presence of faecal droppings ($k = 4$) and presence of trails ($k = 5$).

Human data are collected from $N_h$ households and consist of an infection outcome $Z_{i,j}$ for individual $j$ at household location $i$, for $i = N_r + 1, \ldots, N_r + N_h$, collected at a discrete set of locations $X = \{x_i : i = N_r + 1, ..., N_r + N_h\}$.

Let '[·]' be a shorthand notation for 'the probability distribution of .' We write $Y = (Y_1, \ldots, Y_{N_r})$, $Z = (Z_{N_r+1}, \ldots, Z_{N_r+N_h})$ and $R = (R(x_1), \ldots, R(x_{N_r+N_h}))$. We assume that the $Y_{i,k} : k = 1, ..., 5$ and $Z_{i,j}$ are conditionally independent given $R(x_i)$, from which it follows that

$$[Y, Z|R] = \prod_{i=1}^{N_r} \prod_{k=1}^{5} [Y_{i,k}|R(x_i)] \prod_{i=N_r+1}^{N_r+N_h} \prod_{j=1}^{J_i} [Z_{i,j}|R(x_i)]. \tag{1}$$

where [·] is a shorthand notation for 'the distribution of' and $J_i$ denotes the number of individuals at household $i$. This model structure is shown schematically in *Figure 2*. The conditional independence assumption in *Equation 1* is reasonable for a vector-borne disease or one that is transmitted indirectly, in which context the observed rat indices are to be considered as noisy indicators of the unobservable spatial variation in the extent to which the environment is contaminated with rat-derived pathogen. It would be more questionable for applications in which the disease of interest is spread by direct transmission from rat to human.

### Rattiness

We define rattiness at location $x$ as

$$R(x_i) \quad = \quad d_r^\mathsf{T}(x_i)\beta_r + \sqrt{\psi}\, S(x_i) + \sqrt{1 - \psi}\, U_i. \tag{2}$$

The terms on the right-hand side of *Equation 2* have the following interpretations: $d_r(x_i)$ is a vector of explanatory variables with associated regression coefficients $\beta_r$ is a set of independently and identically distributed zero-mean Gaussian variables with unit variance; $S(x_i)$ is a stationary and isotropic spatial Gaussian process; $\psi \in (0, 1)$ regulates the relative contributions of spatially structured variation, $S(x_i)$, and unstructured random variation, $U_i$, to $R(x_i)$.

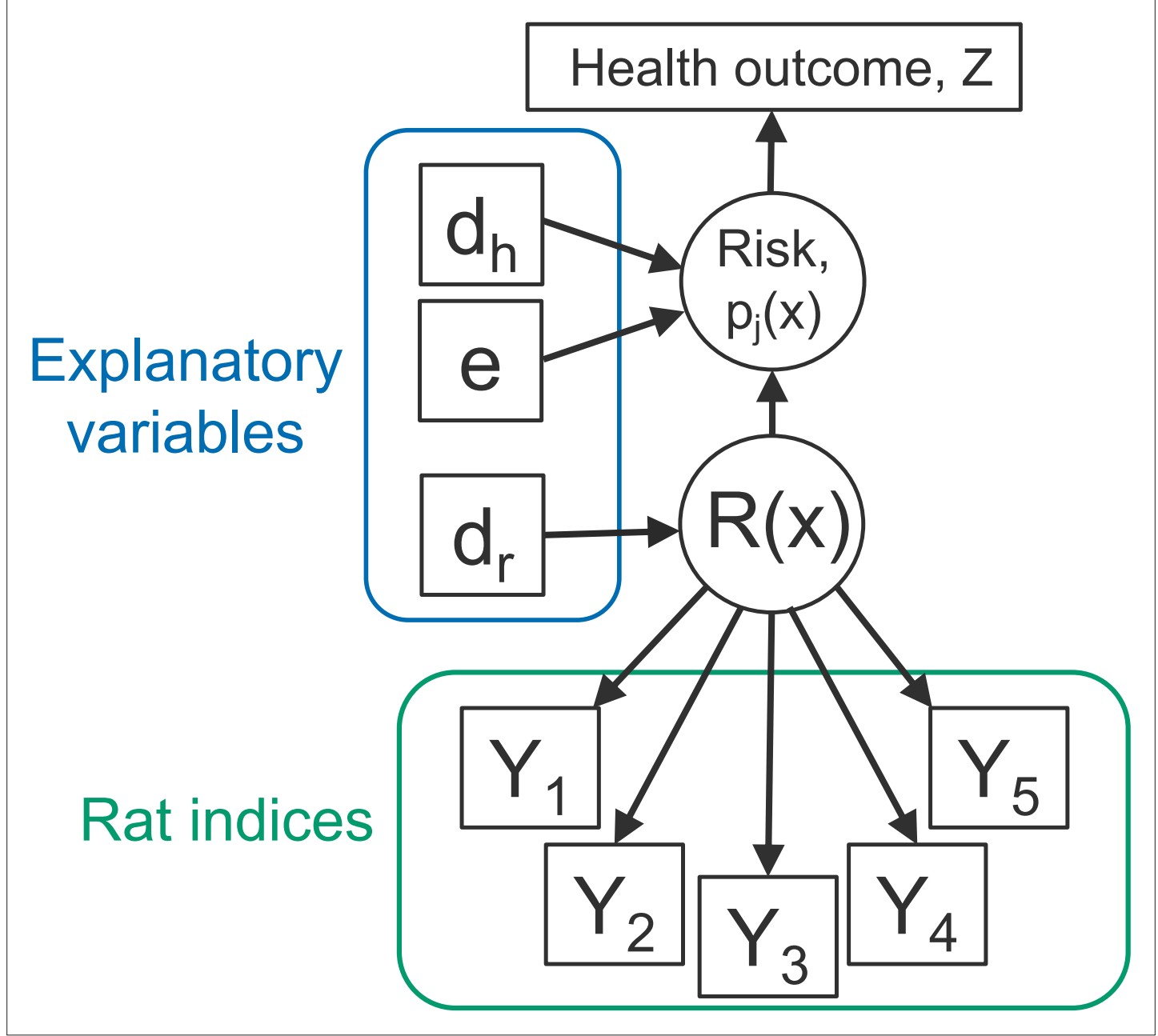

**Figure 2.** Directed acyclic graph (DAG) of the rattiness-infection model framework. $R(x)$ is the value of a spatially continuous stochastic rattiness process at location $x$. The outcome variables $Y_k : k = 1, ..., 5$ are the set of five rat abundance indices that provide information about $R(x)$: traps ($k = 1$), track plates ($k = 2$), number of burrows ($k = 3$), presence of faecal droppings ($k = 4$) and presence of trails ($k = 5$). The outcome variable $Z_{i,j}$ is the observed health outcome, in this case this represents infection status. The terms $d_h$ and $d_r$ represent the sets of spatially continuous explanatory variables which contribute to spatial variation in infection risk in humans and $R(x)$, respectively. The terms $d_h$ and $d_r$ are not mutually exclusive groups of explanatory variables and the same variables may contribute to both infection risk and $R(x)$. The term $e$ represents a set of individual- and household-level explanatory variables which contribute to variation in infection risk. Square objects correspond to observable variables, and circles to latent random variables.

For the Gaussian process, $S(x_i)$, we specify an exponential spatial correlation function:

$\text{Corr}(S(x), S(x)) = e^{-u/\phi}$ where $u = \|x - x'\|$ is the Euclidean distance between $x$ and $x'$, and $\phi$ regulates how fast the spatial correlation decays to zero with increasing distance u.

## Rat abundance outcomes

The variable $Y_{i,1}$, conditionally on $R(x_i)$, is a binomial variable representing the number of traps, out of $n_{i,1}$, in which rats were captured. We assume that the times of rat captures from a trap follow a time-varying inhomogeneous Poisson process with intensity $t_i\mu_1(x_i)$, where $t_i$ is the time (in days) for which a trap is operative and $\log\{\mu_1(x_i)\} = \alpha_1 + \sigma_1 R(x_i)$. It follows that the probability of capturing a rat is

$$1 - \exp\{-t_i\mu_1(x_i)\}.$$

If a trap is found closed without a rat, we assume that the trap was disturbed and set $t = 0.5$. In all other cases, $t = 1$ day. We conducted a sensitivity analysis for this assumption (see 'Appendix 6') and found that it did not materially affect rattiness parameter estimates (*Appendix 6—table 1*).

$Y_{i,2}$, is the number of track-plates, out of $n_{i,2}$, that show presence of rats. We model this as a binomial variable with $n_{i,2}$ trials and probability $\mu_2(x_i)$ where $\log\{\mu_2(x_i)/(1 - \mu_2(x_i))\} = \alpha_2 + \sigma_2 R(x_i)$.

$Y_{i,3}$, is the number of active rat burrows found at location $x_i$. We model this as a Poisson variable with rate $\mu_3(x_i)$ where $\log\{\mu_3(x_i)\} = \alpha_3 + \sigma_3 R(x_i)$.

The variables $Y_{i,4}$ and $Y_{i,5}$ are binary indicators taking value 1, if at least one faecal dropping or trail, respectively, was found at location $x_i$ and 0 otherwise. We model the probability of finding a sign of faecal droppings or trails, $\mu_4(x_i)$ and $\mu_5(x_i)$, using logit-linear regressions $\log\{\mu_4(x_i)/(1 - \mu_4(x_i))\} = \alpha_4 + \sigma_4 R(x_i)$ and $\log\{\mu_5(x_i)/(1 - \mu_5(x_i))\} = \alpha_5 + \sigma_5 R(x_i)$.

## Human infection outcome

Conditionally on $R(x_i)$, we model the binary human infection outcome $Z_{i,j}$ as a Bernoulli variable with the probability, $p_j(x_i)$, that individual $j$ at location $i$ is infected. This is modelled with a logit link function and the following linear predictor

$$\log\left\{\frac{p_j(x_i)}{1-p_j(x_i)}\right\} \quad = \quad \alpha_h + d_h^\mathsf{T}(x_i)\beta_h + e_{i,j}^\mathsf{T}\gamma + \xi(x_i)R(x_i) + V_i \tag{3}$$

where: $d_h(x_i)$ is a vector of spatially continuous explanatory variables with associated regression coefficients $\beta_h$ is a vector of household-level and individual-level explanatory variables with associated regression coefficients $\gamma$; $V_i$ is a set of independently and identically distributed zero-mean Gaussian variables with variance $\sigma^2$ representing unexplained household-level variation; $\xi(x_i)$ regulates the contribution of rattiness to risk of infection.

## Parameterising to test for an interaction with relative elevation

To explore variation in the role of local rat populations in transmission within sections of the study area with different flooding risk profiles, $\xi$ was parameterised to test for an interaction between rattiness and a categorical parameterisation of household elevation relative to the bottom of the valley (modelled as a piecewise constant function with breaks at 6.7 and 15.6 m, resulting in three categories: low, medium and high elevation levels.) on human infection risk. This was implemented by first dividing the study area into three elevation categories with different flooding risk profiles (as observed during our work in the study area over the last 15 years): low ($0 - 6.7m$ from bottom of valley; high flooding risk with maintenance of floodwater for long periods), medium ($6.7 - 15.6m$; moderate flooding with high water runoff), and high ($> 15.6m$; limited flooding and water runoff). Our study was then designed to evenly sample across this elevation gradient and minimum and maximum values for each elevation category were chosen to include an equal number of households in each level. We then define the set of household locations in each low, medium, and high elevation category as $x_{low}$, $x_{med}$, and $x_{high}$, respectively. Three values of $\xi$ were then estimated such that:

$$\xi(x_i) = \begin{cases} \xi_{low} & \text{at locations } x_i \in x_{low} \\ \xi_{med} & \text{at locations } x_i \in x_{med} \\ \xi_{high} & \text{at locations } x_i \in x_{high} \end{cases} \tag{4}$$

## Variable selection

### Predictors of rattiness

The exploratory analysis for the rattiness model followed the steps developed and described previously (*Eyre et al., 2020*). Firstly, we explored the functional form of the relationship between rattiness and three continuous explanatory variables: relative elevation, distance to large refuse piles and land cover type. To do this, we fitted a simplified rattiness model that did not include covariates or account for spatial correlation. Rattiness is consequently modelled purely as unstructured random variation; hence $R(x_i) = U_i$ (*Eyre et al., 2020*). We then computed the predictive expectation of this simplified rattiness process, $\hat{U}_i$, at all locations for which rat index measurements were observed. A generalized additive model (GAM) (*Hastie and Tibshirani, 1987*) was then fitted to the $\hat{U}_i$ with the three explanatory variables and the shape of each fitted smooth function was used to assess whether the relationship between each variable and rattiness was linear. Non-linear relationships were modelled using linear splines based on the identified functional form, with knots placed at relative elevations of 8 m and 22 m, and at a distance from large refuse piles of 50 m (see *Appendix 1—figure 1*). For variable selection, linear models with all combinations of these variables were fitted and ranked by their Akaike Information Criterion (AIC) value (*Bozdogan, 1987*). The model with the lowest AIC included all of the variables and their linear splines (*Appendix 2—table 1*).

Following the methodology outlined previously (*Eyre et al., 2020*), we fitted the full geostatistical rattiness model using the variables selected in 'Predictors of rattiness'. We then plugged in the maximum likelihood estimates and made predictions for rattiness at all human household locations; here, the predictive target is $T(x) = d_r(x)^{\mathsf{T}}\beta_r + \sqrt{\psi}S(x)$ rather than $R(x)$ as defined by *Equation 2* because the predicted value of the spatially uncorrelated $U(x)$ at any location $x$ where rat abundance indices have not been recorded is zero. The expectation of this predictive distribution was then computed to provide an estimate of mean predicted rattiness at all household locations. This was then used as an exploratory covariate in the following section.

### Risk factors for human infection

All explanatory variables were grouped into the following four domains: social status, household environment, occupational exposures and behavioural exposures (see Table 2 for the full list of considered variables by group). A group of a priori confounding variables was then identified, with age, gender and household per capita income selected based on previous findings (*Hagan et al., 2016*; *Reis et al., 2008*; *Felzemburgh et al., 2014*), and valley also included to account for otherwise unmeasured differences between the three valley regions within the study area. In the household environment domain, two variables were used to capture risk due to sewer flooding close to the household: (i) the presence of an open sewer within 10 metres of the household location and (ii) a binary 'unprotected from open sewer' variable which identified those households within 10 metres of an open sewer that did not have any physical barriers erected to prevent water overflow. Three high-risk occupations were included in the occupational exposures domain as binary variables. Construction workers and refuse collectors have direct contact with potentially contaminated soil, building materials and refuse in areas that provide harbourage and food for rats. Travelling salespeople have regular and high levels of exposure to the environment (particularly during flooding events) as they move from house to house by foot. Two other binary occupational exposure variables were included that measured whether a participant worked in an occupation that involves contact with floodwater or sewer water.

The relationship between continuous explanatory variables and infection risk (on the log-odds scale) was assessed for linearity by fitting a GAM while controlling for the four confounders. As before, non-linear relationships were modelled using linear splines based on the identified functional form. Age was modelled with a knot at 30 years old, education at 5 years and relative elevation at 20 m (*Appendix 1—figure 2*). A univariable analysis was conducted to explore the relationship between each explanatory variable and infection risk while controlling for the four a priori confounding variables. Crude and adjusted odds ratios were estimated using a mixed effects logistic regression with a random effect to account for unexplained variation at the household-level.

For the multivariable model, variable selection was conducted within each domain separately. Mixed-effect logistic regression models were fitted for all combinations of the variables in each domain and were ranked by their Akaike Information Criterion (AIC) value (*Appendix 2—table 2*). Variables in the model with the minimum AIC value were selected for each domain. Age, gender, household per

capita income and valley were controlled for in all models throughout this process. Then, the variables selected from each domain were combined and the mean predicted rattiness estimate (obtained in 'Predictors of rattiness at each household location') was included with an interaction with relative elevation category. This set of variables was reduced once more following the same process and all selected variables were included in the final multivariable model ('Appendix 3').

## Model fitting

All rat and human variables selected in 'Variable selection' were then included in the full joint model defined in *Equation 2* and *Equation 3*. We fit this model using the Monte Carlo maximum likelihood (MCML) method (*Christensen, 2004*) as described in 'Appendix 4', and compute 95% confidence intervals by re-fitting the model for 1000 parametric bootstraps. A formal diagnostic investigation of randomized quantile residuals (*Dunn and Smyth, 1996*; *Smyth et al., 2021*) is included 'Appendix 7'. We found no evidence in the diagnostic plots (*Appendix 7—figure 1*) to suggest that there were issues with our modelling approach.

## Prediction maps

The maximum likelihood parameter estimates were then used to make prediction maps for rattiness and infection risk as follows.

To map a general predictive target, $T(x)$ say, we first define $T^* = (T(x_1^*), \ldots, T(x_H^*))$, where $X^* = \{x_1^*, \ldots, x_H^*\}$ is a finely spaced grid of locations to cover the region of interest. We then draw samples from the predictive distribution of $T^*$, that isits conditional distribution given all relevant data. These samples can then be used to compute any desired summary of the predictive distribution. In our analysis, we used as summaries the expectation and 95% prediction interval.

Our first predictive target is rattiness, for which $T(x) = d_r^\mathsf{T}(x)\beta_r + \sqrt{\psi}S(x)$. Our second is human infection risk, for which $T(x) = d_h^\mathsf{T}(x)\beta_h + e^\mathsf{T}\gamma + \xi(x_i)R(x) + V_i$. In either case, we first sample from $[R|W; \theta, \omega]$ using the same sampling algorithm as for maximizing the likelihood in 'Joint modelling rat abundance and human infection: the rattiness-infection framework', with the parameters θ and $\omega$ fixed at their maximum likelihood estimates. After obtaining samples $r_{(b)}$, $b = 1, \ldots, B$, we then sample from $[T^*|r_{(b)}]$, which in both cases follows a multivariate Gaussian distribution with mean and covariance matrix easily obtained from their joint Gaussian distribution, $[R, T^*]$. The resulting values, $t_{(b)}(x_h^*), h = 1, \ldots, H; b = 1, \ldots, B$, constitute $b$ samples drawn from $[T^*|W]$ as required. Note that each $t_{(b)}(x_h^*), h = 1, \ldots, H$ is a sample from the joint predictive distribution of the complete surface of $T(x)$ over the whole of the region of interest and can therefore be used to make inferences about spatially aggregated properties of $T(x)$ if required.

## Data and code accessibility

Data and code used in this analysis are publicly available at https://github.com/maxeyre/Rattiness-infection-framework, (copy archived at swh:1:rev:e7953d38269ce97221dbdd83c0be2c65d92dff40, *Eyre, 2022*) and have been published (*Eyre et al., 2021*). However, household coordinates and valley ID have been removed from the human data to ensure participant anonymity. The analysis was conducted using R (*R Development Core Team, 2016*) and the following packages: tidyverse (*Wickham, 2017*), mgcv (*Wood and Wood, 2015*), PrevMap (*Giorgi and Diggle, 2017*), MuMIn (*Barton, 2020*), lme4 (*Bates et al., 2007*), and statmod (*Smyth et al., 2021*). We also include a step-by-step explanation of the model building process to guide future users of the rattiness-infection framework in 'Appendix 8'.

# Results

## Study overview

In Pau da Lima, we identified 3179 eligible residents using a baseline community census, household visits and through other members of the household. Of these, 2018 (63.4%) individuals consented to join the study and provided a blood sample in the first serosurvey (August-October 2014). As a result of loss to follow-up, only 1401 (69.4%) of these participants (from 669 households) completed the second serosurvey (January-April 2015). Individuals were lost to follow-up because they could not be found after at least five attempts (44.4%), had moved out of the study area (31.1%) or did not

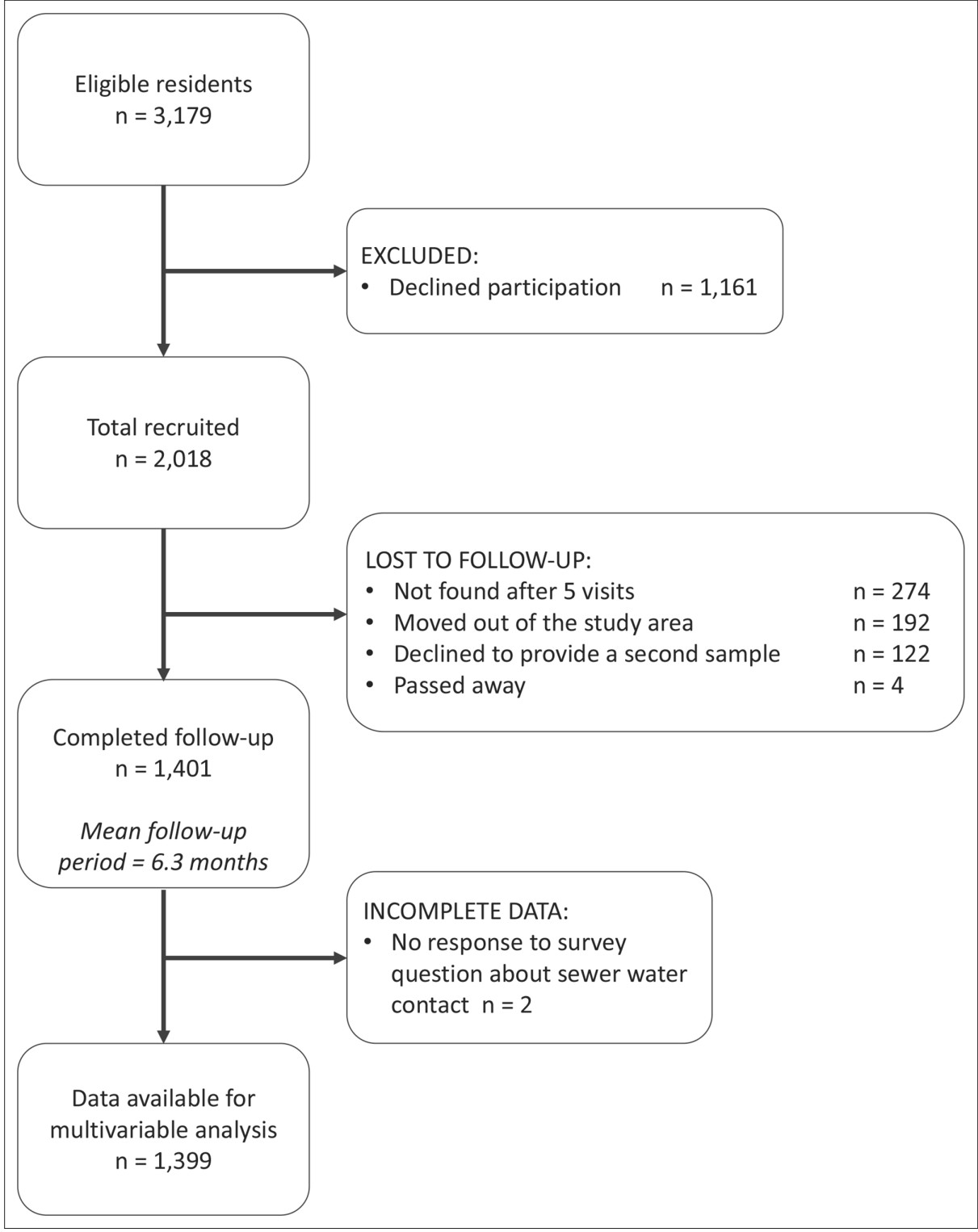

**Figure 3.** The study participant flow chart in line with the STROBE (Strengthening the Reporting of Observational Studies in Epidemiology) statement (http://www.strobestatement.org).

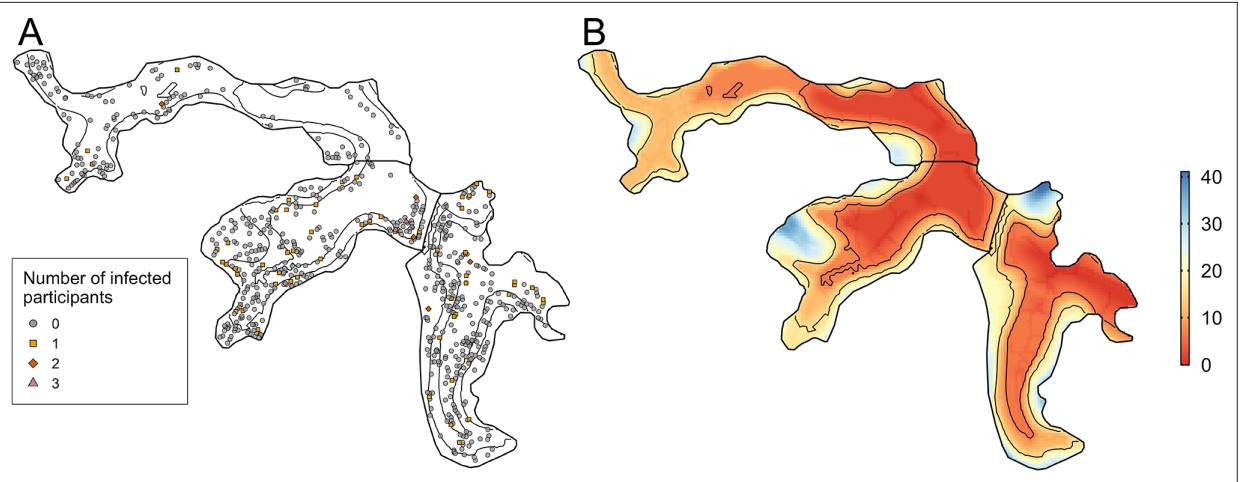

**Figure 4.** Household infection and elevation maps. (**A**) Map of participant household locations with the number of leptospiral infections in each household marked (grey circle - no infections; orange square - 1 infection; red diamond - 2 infections; pink triangle - 3 infections) and contours marking low, medium, and high relative elevation category; (**B**) Elevation (metres) relative to the bottom of the valley with contours marking low, medium, and high relative elevation levels.

wish to provide a second blood sample (19.8%). An overview of participant recruitment is provided in *Figure 3*. Individuals lost to follow-up were similar in age to those who remained in the study cohort (mean 29.0 and 28.8 years old, respectively, t-value = −0.37, $df = 1288.5$, $p = 0.7$) but were more likely to be male (49.8% male compared to 42.6%, $\chi^2 = 8.5$, $df = 1$, $p < 0.01$). A full description of the study cohort is included in *Appendix 5—table 1*.

Between the two serosurveys there was serological evidence of 72 leptospiral infections in the cohort, with an overall infection rate of 51.4 (95%CI 40.4, 64.2) infections per 1,000 follow-up events. Valleys 2 and 3 had high estimated infection rates with 66.4 (95%CI 47.3, 90.2) infections per 1000 follow-up events and 49.6 (95%CI 33.6, 69.9) infections per 1000 follow-up events, respectively, compared to 23.2 (95%CI 9.2, 46.9) infections per 1000 follow-up events in Valley 1. The number of infected participants in each household are mapped in *Figure 4* - panel A, with relative elevation shown for reference in *Figure 4* - panel B.

In the rat ecology study a rat was captured in 129 (9.0%) out of 1,512 trapping-days, 263 (37.4%) out of 703 track plate days had at least one positive plate and 28.5%, 19.7%, and 25.9% of the 580 sampled locations had at least one sign of active burrows, faecal droppings and trails, respectively.

## Exploratory analysis and model selection

The results from the exploratory multivariable analysis of rattiness are shown in *Table 1*. The linear splines used were informed by the functional forms shown in *Appendix 1—figure 1*. The relationship between rattiness and relative elevation demonstrates a trade-off between the high availability of food sources at the bottom of the valley and high risk of flooding which prevents the establishment of burrows. In the lowest elevation

**Table 1.** Multivariable linear regression analysis of predictors for rattiness (note that rattiness is a unit-variance random variable when interpreting the magnitude of effect estimates).

| Variable | Estimate (95% CI) * |
|---|---|
| Relative elevation (per 1 m increase)[†] | |
| 0–8 m | 0.04 (0.00, 0.07) |
| 8–22 m | −0.04 (-0.09, 0.01) |
| >22 m | 0.06 (0.00, 0.10) |
| Distance to large refuse piles (per 10 m increase)[†] | |
| 0–50 m | −0.07 (-0.13,−0.01) |
| >50 m | 0.02 (-0.05, 0.09) |
| Impervious land cover (per 10% increase) | −0.05 (-0.08,−0.01) |

*CI, Confidence interval.

[†]The effects of relative elevation and distance to refuse are modelled as broken linear models with transitions at 8m and 22m, and 50m, respectively. This was informed by the relationship described by Generalized Additive Modelling in *Appendix 1—figure 1*.

areas (0–8 m above the bottom of the valley), relative elevation and rattiness were positively associated with an increase of 0.04 (95%CI 0.00, 0.07) rattiness units per 1 m; when interpreting the magnitude of effect estimates note that, by definition, rattiness is defined so as to have variance one. Rattiness then peaked at an elevation of 8 m before declining with increasing elevation by 0.04 (95%CI –0.09, 0.01) units until an elevation of 22 m. Rattiness started to increase again above this elevation by 0.06 (95%CI 0.00, 0.10) units per metre. Rattiness decreased with increasing distance from large refuse piles, a source of food and harbourage, by 0.07 (95%CI –0.13,–0.01) units per 10 m distance until a distance of 50 m, beyond which there was a smaller increase in rattiness of 0.02 (95%CI –0.05, 0.09) per 10 m. Impervious land cover (defined as the proportion of the area within a 30 m radius around each sampling location classified as pavement or building) was negatively associated with rattiness, decreasing by –0.05 (95%CI –0.08,–0.01) units for every 10% increase in impervious cover.

In the community cohort data, the univariable analysis identified several risk factors that increased a resident's risk of leptospiral infection (*Table 2*). Variables in two of the four domains (demographic and social status and behavioural exposures) had estimated effect sizes with 95% confidence intervals that did not include an odds ratio of one (statistically significant at the conventional 5% level). Within the demographic and social status domain, risk of infection increased with age and was found to be higher for male participants and those living in Valleys 2 and 3. In the behavioural exposures domain, participants who had had frequent contact with floodwater in the last six months were more likely to be infected. Two individuals were excluded from the multivariable analysis (n=1399) because of missing data for the floodwater exposure survey question.

In the exploratory results from the multivariable model there was strong evidence of an interaction between rattiness and household relative elevation category on human infection risk (see *Appendix 3—table 1* for all parameter estimates for this model). In the high elevation category area, a unit increase in mean predicted rattiness at the household location was estimated to increase the odds of infection by 6.92 (95%CI 1.88, 25.47). In contrast, in the low and medium elevation category areas there was no evidence of a relationship between rattiness and infection risk, as shown in *Figure 5*. Consequently, this interaction effect was also included in the rattiness-infection joint model.

The explanatory variables selected in the rat and human multivariable analyses were then entered into the full rattiness-infection joint model with the functional forms included in *Table 1*. To test for residual spatial correlation in the human infection data after controlling for explanatory variables and rattiness, we fitted the joint model with an additional spatial Gaussian process in the human infection linear predictor. The estimated value for the scale of spatial correlation for this Gaussian process was less than 1 m and indistinguishable from household-level variation. We consequently fitted the joint model specified in *Equation 3* which assumes that there is no residual spatial correlation in the human infection data.

## Joint rattiness-infection model

Human infection risk factors, rattiness predictors and other model parameters estimated using the joint rattiness-infection model are shown in *Table 3*. Infection risk was strongly associated with age, with an individual experiencing an increased odds of infection of 1.09 (95%CI 1.04, 1.19) for every year of life up until 30 years of age, and 1.02 (95%CI 0.92, 1.09) for each additional year thereafter. Male participants were more likely to be infected than female participants (OR 2.69 95% CI 1.58, 5.89). Compared with individuals living in Valley 1, those living in Valley 2 had a higher estimated odds of infection (OR 2.91 95% CI 1.03, 20.82). Individuals living in the medium (OR 0.77 95% CI 0.31, 1.66) and high (OR 0.67 95% CI 0.11, 1.64) elevation areas had a lower estimated odds of infection relative to those living in the low relative elevation category area where there are open sewers and flooding risk is higher, however these confidence intervals included an odds ratio of one (not statistically significant at the conventional 5% level).

Infection risk was positively associated with rattiness for households situated in all three levels of the relative elevation category variable. However, while the effect size (per unit increase in rattiness) was similar in the low (OR 1.14 95% CI 1.05, 1.53) and medium (OR 1.25 95% CI 1.08, 1.74) elevation areas, in the high elevation area the effect of increasing rattiness on infection risk was significantly stronger (OR 3.27 95% CI 1.68, 19.07). This interaction effect between rattiness and household relative elevation category on human infection risk was confirmed with a test for evidence against the null hypothesis that $\xi_{low} = \xi_{med} = \xi_{high}$ ($p = 0.026$, $\chi^2 = 7.33$, $df = 2$).

Table 2. Univariable mixed effects logistic regression analysis of human risk factors for leptospiral infection.

| Variable | OR (95% CI)* | aOR (95% CI)* |
|---|---|---|
| **Demographic and social status** | | |
| Age (per year)[†] | | |
| 0–30 years old | 1.08 (1.03, 1.13) | 1.09 (1.04, 1.15) |
| >30 years old | 1.02 (0.96, 1.09) | 1.02 (0.95, 1.08) |
| Male gender | 2.22 (1.31, 3.85) | 2.78 (1.56, 4.96) |
| Daily per capita household income (US$/day) | 1.01 (0.89, 1.11) | 0.92 (0.80, 1.05) |
| Valley | | |
| 1 | REF | REF |
| 2 | 3.35 (1.33, 10.37) | 3.52 (1.23, 10.05) |
| 3 | 2.39 (0.93, 7.38) | 2.53 (0.88, 7.27) |
| Adult illiteracy | 1.34 (0.61, 2.79) | 0.66 (0.29, 1.49) |
| Education (per year of education)[†] | | |
| 0–5 years | 1.05 (0.85, 1.32) | 1.14 (0.91, 1.44) |
| >5 years | 0.96 (0.73, 1.27) | 0.96 (0.75, 1.26) |
| **Household environment** | | |
| Impervious land cover (per 10% increase) | 0.87 (0.76, 0.99) | 0.82 (0.71, 0.95) |
| Relative elevation (per 1 m increase)[†] | | |
| 0–20 m | 0.94 (0.89, 0.99) | 0.93 (0.88, 0.99) |
| >20 m | 1.12 (0.98, 1.29) | 1.12 (0.97, 1.29) |
| Relative elevation category [‡] | | |
| Low (0–6.7 m) | REF | REF |
| Medium (6.7–15.6 m) | 0.72 (0.37, 1.39) | 0.72 (0.36, 1.44) |
| High (>15.6 m) | 0.58 (0.27, 1.20) | 0.51 (0.23, 1.11) |
| Open sewer within 10 m | 1.60 (0.85, 3.17) | 1.69 (0.85, 3.37) |
| Unprotected from open sewer | 1.00 (0.55, 1.79) | 1.11 (0.61, 2.03) |
| Live on hillside | 0.99 (0.52, 1.86) | 0.89 (0.46, 1.71) |
| **Occupational exposures** | | |
| Work in construction [§] | 1.36 (0.51, 3.21) | 0.62 (0.23, 1.67) |
| Work as travelling salesperson [§] | 4.81 (1.12, 18.78) | 2.97 (0.71, 12.40) |
| Work in refuse collection [§] | 2.95 (1.04, 7.89) | 1.57 (0.56, 4.42) |
| Work involves contact with floodwater [§] | 0.89 (0.04, 5.61) | 0.52 (0.05, 4.96) |
| Work involves contact with sewer water [§] | 3.61 (0.45, 20.38) | 1.92 (0.29, 12.80) |
| **Behavioural exposures** | | |
| Contact with floodwater in last 6 months | | |
| Never/rarely | REF | REF |
| Sometimes | 0.61 (0.27, 1.25) | 0.66 (0.30, 1.47) |
| Frequently | 2.14 (0.91, 4.94) | 2.84 (1.18, 6.86) |
| Contact with sewer water in last 6 months | | |

*Table 2 continued on next page*

*Table 2 continued*

| Variable | OR (95% CI)* | aOR (95% CI)* |
|---|---|---|
| Never/rarely | REF | REF |
| Sometimes | 0.55 (0.19, 1.31) | 0.67 (0.25, 1.78) |
| Frequently | 1.42 (0.51, 3.50) | 1.63 (0.61, 4.41) |

*OR, Odds ratio; aOR, Adjusted odds ratio; CI, Confidence interval; REF, Reference level.

†The effect of age, education and relative elevation are modelled as broken linear models with transitions at 30 years old, 5 years of education and an elevation of 20m. This was informed by the relationship described by Generalized Additive Modelling (*Appendix 1—figure 2*).

‡Relative elevation category consists of three discrete groups representing three regions with different floodingrisk profiles.

§Binary variable with reference category of 'no occupational exposure'.

Parameter estimates for the rattiness variables were very similar to the estimates from the exploratory linear regression (*Table 1*), with a slightly higher effect size for the distance to refuse piles and land cover variables. There was evidence of small-scale spatial correlation in rattiness ($\phi$ = 9.23m 95% CI 3.21, 18.24 m) corresponding to a spatial correlation range (the distance at which the correlation reduces to 5%) of approximately 28 m. The estimate for $\psi$ of about 0.67 (95%CI 0.29, 1.00)

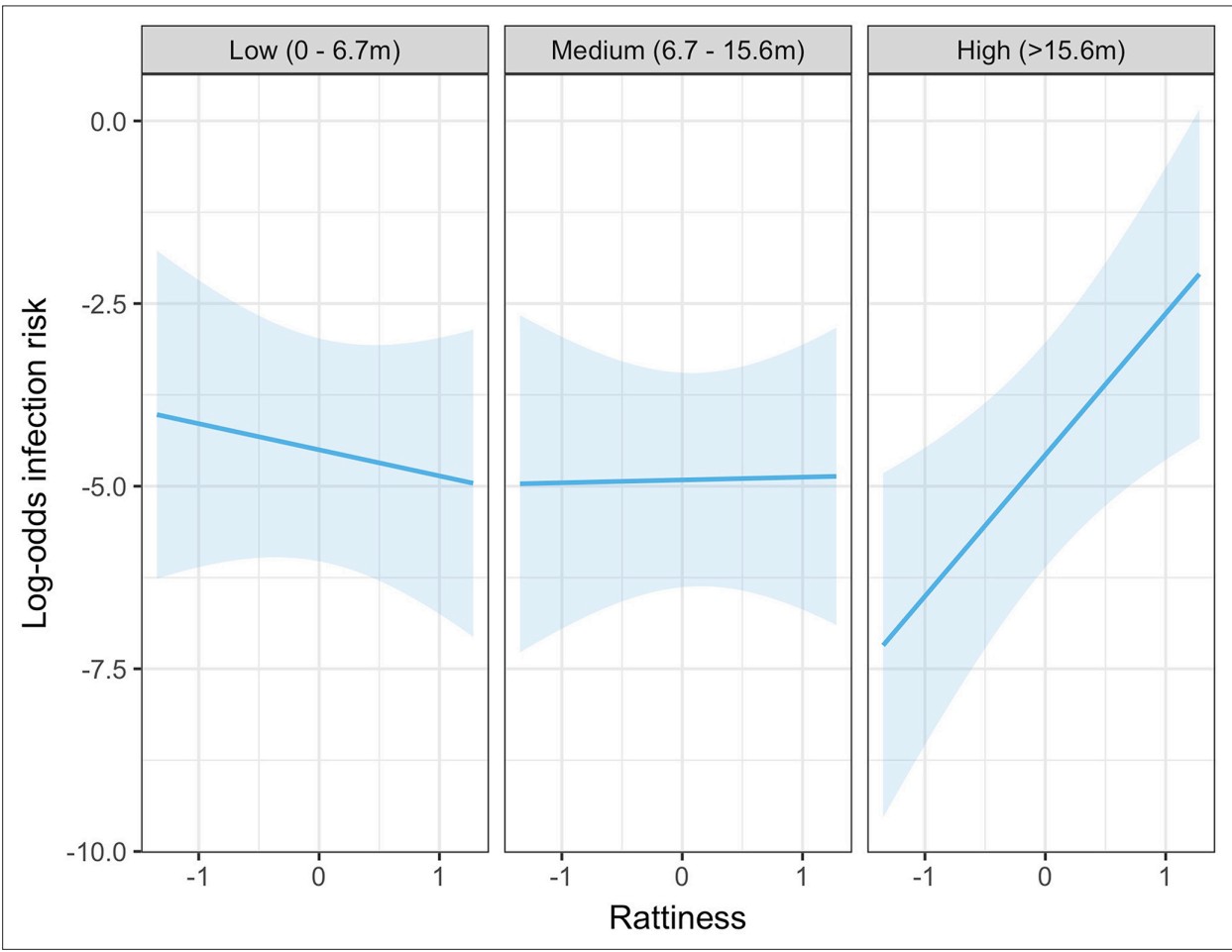

**Figure 5.** Predicted relationship between rattiness and infection risk from the multivariable mixed effects logistic regression demonstrating evidence of an interaction with relative elevation category (low, medium and high). Shown on the log-odds scale with shaded areas corresponding to 95% confidence intervals.

**Table 3.** Parameter estimates for the full joint rattiness-infection model.

| Parameter | Estimate (95% CI) |
|---|---|
| Human infection risk factors | OR |
| Age (per year) | |
| 0–30 years old | 1.09 (1.04, 1.19) |
| >30 years old | 1.02 (0.92, 1.09) |
| Male gender | 2.69 (1.58, 5.89) |
| Daily per capita household income (US$/day) | 0.93 (0.74, 1.05) |
| Valley | |
| 1 | REF |
| 2 | 2.91 (1.03, 20.82) |
| 3 | 2.28 (0.86, 14.00) |
| Relative elevation category | |
| Low (0–6.7 m) | REF |
| Medium (6.7–15.6 m) | 0.77 (0.31, 1.66) |
| High (>15.6 m) | 0.67 (0.11, 1.64) |
| Work as travelling salesperson | 3.16 (0.38, 20.57) |
| Contact with floodwater in last 6 months | |
| Never/rarely | REF |
| Sometimes | 0.62 (0.18, 1.39) |
| Frequently | 2.47 (0.67, 7.41) |
| Rattiness (per unit rattiness) | |
| $\xi_{low}$ | 1.14 (1.05, 1.53) |
| $\xi_{med}$ | 1.25 (1.08, 1.74) |
| $\xi_{high}$ | 3.27 (1.68, 19.07) |
| $\sigma^2$ (variance of household-level random effect) | 1.36 (0.23, 5.35) |
| Rattiness variables | |
| Relative elevation (per 1 m increase)[2] | |
| 0–8 m | 0.05 (-0.01, 0.13) |

*Table 3 continued on next page*

*Table 3 continued*

| Parameter | Estimate (95% CI) |
|---|---|
| 8–22 m | –0.06 (-0.16, 0.02) |
| >22 m | 0.05 (-0.03, 0.14) |
| Distance to large refuse piles (per 10 m increase)[3] | |
| 0–50 m | –0.10 (-0.21, 0.02) |
| >50 m | 0.03 (-0.11, 0.17) |
| Impervious land cover (per 10% increase) | –0.07 (-0.14,–0.01) |
| Rattiness parameters | |
| $\alpha_{traps}$ | –2.94 (-3.27,–2.65) |
| $\alpha_{plates}$ | –2.06 (-2.50,–1.74) |
| $\alpha_{burrows}$ | –1.41 (-1.67,–1.16) |
| $\alpha_{faeces}$ | –2.82 (-3.83,–2.32) |
| $\alpha_{trails}$ | –2.22 (-2.96,–1.76) |
| $\sigma_{traps}$ | 0.72 (0.45, 0.97) |
| $\sigma_{plates}$ | 2.37 (2.05, 2.68) |
| $\sigma_{burrows}$ | 1.28 (1.08, 1.45) |
| $\sigma_{faeces}$ | 2.36 (1.80, 3.34) |
| $\sigma_{trails}$ | 2.43 (1.85, 3.12) |
| $\psi$ | 0.67 (0.29, 1.00) |
| $\phi$ | 9.23 (3.21, 18.24) |

indicates that the majority of the unexplained variation in rattiness is spatially structured, with the remainder modelled as a nugget effect.

## Spatial prediction

There was heterogeneous spatial variation in predicted rattiness. The numerous small regions of high rattiness in *Figure 6* - panel A are indicative of the small-scale spatial correlation in the data. The low elevation areas in the central length of each valley (relative elevation is shown

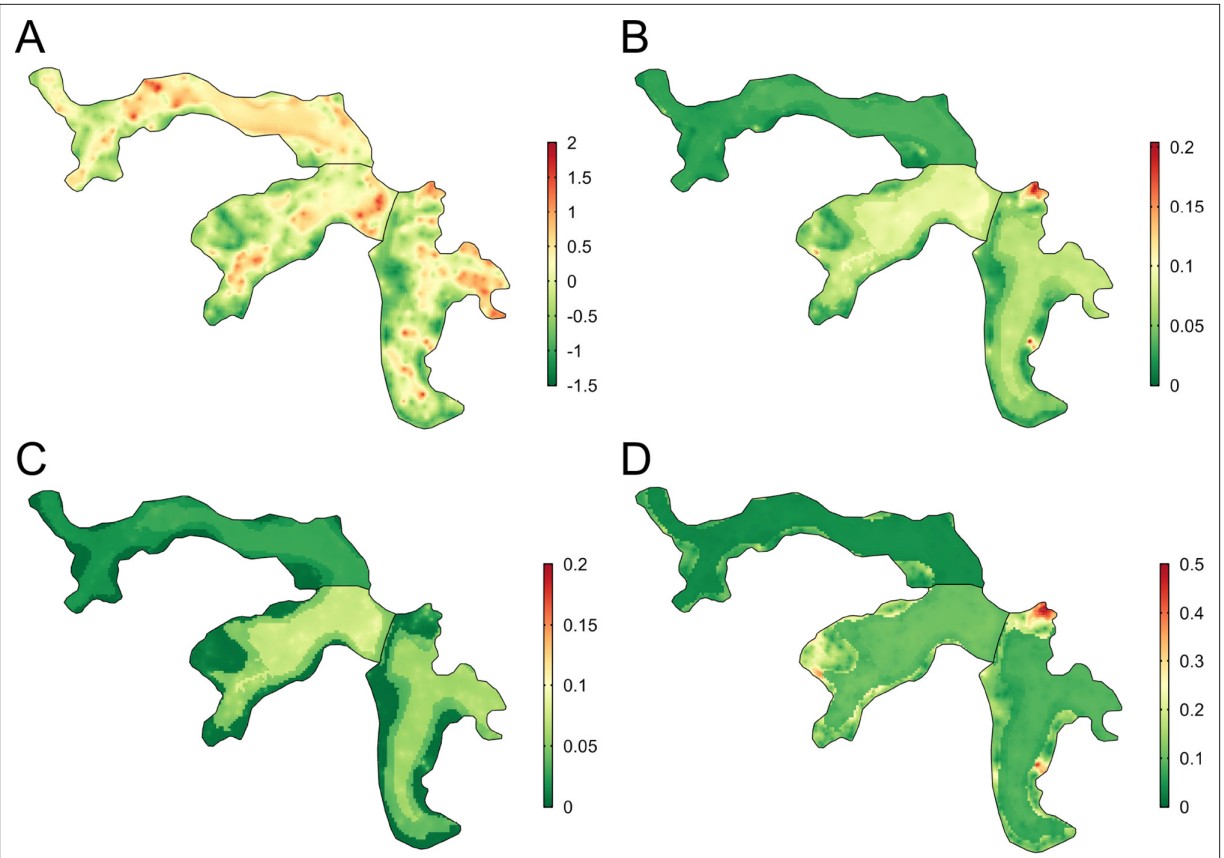

**Figure 6.** Joint rattiness-infection model predictions. (**A**) Mean predicted rattiness; (**B**) Mean predicted leptospiral infection risk for 30-year-old male participants with a household per capita income of USD$1 /day who never/rarely have contact with floodwater and do not work as a travelling salesperson; (**C**) lower 95% prediction interval for predicted infection risk; (**D**) upper 95% prediction interval for predicted infection risk.

in *Figure 6* - panel E with the contours marking the low, medium and high elevation areas) had high mean predicted rattiness. High rattiness was also predicted in several high elevation areas, for example the northern tip of Valley 3 and several small hotspots along the three valley's high elevation sides.

To illustrate the spatial variation in infection risk within the study area, prediction maps are shown in *Figure 6* - panel B for a 30-year-old male participant with a household per capita income of USD$1 / day who never or rarely had contact with floodwater in the previous six months and did not work as a travelling salesperson. Infection risk was low across most of Valley 1 (<2.5%), with marginally higher average values found in the central low elevation area (2.5–5%). Risk was consistently higher across most of Valleys 2 and 3 (7.5–15%), with the effect of elevation on risk clearly visible. In areas with higher and more spatially heterogeneous predicted infection risk, for example in the central region of Valley 2, this was driven by high levels of predicted rattiness. The stronger estimated effect of rattiness on infection risk in higher elevation areas was particularly visible in Valleys 2 and 3, as seen in the three hotspots with risk reaching 20% and the moderate risk hotspots along the sides of both valleys. Prediction intervals were relatively narrow across most of the study area (*Figure 6* - panel C and *Figure 6* - panel D) with greater uncertainty in the high risk areas.

## Discussion

We developed and applied a novel framework for joint spatial modelling of disease reservoir abundance and human infection risk to a community-based cohort study and fine-scale rat ecology study. We found that higher levels of rattiness, our proxy for rat abundance, at the household location were associated with a higher risk of leptospiral infection for residents across the entire study area.

Importantly, we found that a unit increase in rattiness in high elevation areas was associated with an almost three times higher odds ratio for infection than in low and medium elevation areas. To our knowledge, this is the first study to jointly model rodent abundance and human infection data for a rodent-borne zoonosis. The findings provide new insights into how the dominant mechanisms of *Leptospira* transmission within complex urban settings may vary over small distances, as a result of interactions between rats, the environment, geography, and local epidemiology.

The finding that rattiness was associated with infection risk indicates that the spatial distribution of rat populations was an important driver of transmission close to the household across the entire study area. This is consistent with a recent study investigating the predictive power of household rat infestation scores for human infection (*Costa et al., 2021*). There was no residual spatial correlation in the infection data after accounting for rattiness in our analysis, possibly suggesting that previously unexplained spatial heterogeneity in risk could be driven by variation in rattiness (*Hagan et al., 2016*). Our model also predicted high average rattiness across the low elevation areas where leptospiral transmission is high (*Hagan et al., 2016*; *Reis et al., 2008*). This supports the hypothesis that abundant rat populations are responsible for high levels of observed environmental contamination across these lower areas (*Casanovas-Massana et al., 2018*; *Casanovas-Massana et al., 2022*), and consequently increased infection risk.

The identified interaction between elevation and rattiness on infection risk suggests that relatively small changes in environment and topography can modify transmission pathways within an urban community. The weaker effect of rattiness on infection risk at low and medium elevation areas relative to high elevation areas may be explained by differences in their hydrological profiles. While high rat abundance in low and medium areas results in high leptospiral contamination, these areas are prone to high levels of water runoff and flooding. This disperses the pathogen across low elevation areas. The ability of leptospires to persist in the environment for weeks or months means that this process can significantly increase environmental risk in low elevation areas for long periods. This process disconnects shedding and infection events in space and time (*Plowright et al., 2017*) and obscures the relationship between infection risk and rattiness in low and medium elevation households.

In contrast, high elevation areas have lower levels of water runoff and flooding due to improved drainage and sewage systems, and a smaller upstream catchment area for rainfall. Leptospires are consequently less likely to be washed away from the location at which they were shed, and environmental risk remains more localised and strongly associated with the spatial distribution of rats. This hypothesised role of hydrology in the aggregation and dispersal of leptospires (*Plowright et al., 2017*) is supported by a recent study in low elevation areas of Pau da Lima which found that soil contamination was not associated with local rat activity (*Schneider et al., 2018*). However, our finding that rattiness was associated with infection in low and medium elevation areas suggests that the spatial distribution of environmental risk in these areas is not entirely determined by water dispersal.

Interestingly, a previous study of surface waters in Pau da Lima found that the probability of a sample being positive for *Leptospira* was highest in low elevation areas and lowest in medium elevation areas, with no significant difference between low and high elevation areas (*Casanovas-Massana et al., 2018*). This is consistent with our findings and suggests that there may be a 'washing out' of locally deposited leptospires in medium elevation areas but not in high elevation areas.

This has several implications for disease control strategies which aim to reduce environmental risk. Improving drainage systems at all elevation levels can reduce the dispersal of leptospires from high to low elevation areas. Closure of sewer systems, which generally run through low elevation areas, can protect local residents from exposure and reduce the introduction of additional contamination from upstream sewer water. Paving over soil surfaces can reduce the surface area over which leptospires can persist (*Bierque et al., 2020*), reducing environmental risk further.

A reduction in the dispersal and accumulation of bacteria will result in more localised environmental risk, as was observed in the high elevation areas in this study. Higher risk will then be found in areas with a high abundance of infected rats. This may also reduce environmental exposure for rats, thereby lowering shedding rates and acting as a feedback loop into the *Leptospira* transmission cycle (*Minter et al., 2018*). Given the limited and short-term impact of chemical rodenticide campaigns on Norway rat abundance in these settings (*de Masi et al., 2009*), longer-term environment management strategies targeted at rattiness hotspots may also be needed to reduce the availability of key predictors of rattiness, such as large refuse piles and vegetation and soil land cover. Funding and political will for

large-scale infrastructural interventions is often limited in marginalised urban settings and small-scale community-based interventions which target these mechanisms should be evaluated.

Transmission is dynamic in space and time and the alignment of conditions which enable spill-over infection can vary over time (*Plowright et al., 2017*). Our study was designed to explore the spatial variation in rattiness and infection risk in Pau da Lima during the driest period of the year, and it may not be representative of transmission mechanisms during the rainy season. There is some evidence, however, that this may not necessarily be the case, with two recent studies in Pau da Lima reporting low seasonal variation in both rat abundance (*Panti-May et al., 2016*) and spatial infection risk patterns (*Hagan et al., 2016*). Nonetheless, future studies across different time periods are needed to establish the role of rat abundance in *Leptospira* transmission.

In this study we used household location to link rattiness to an individual, under the assumption that the majority of their exposure occurs close to home. Given the spatially heterogeneous distribution of rattiness and environmental contamination (*Casanovas-Massana et al., 2018*) within the community, future epidemiological studies of leptospirosis and zoonotic spillover could benefit from trying to pinpoint key sources of infection away from the household using GPS mobility data, as has been attempted in a small study previously (*Owers et al., 2018*). The rattiness-infection framework could then be extended to model cumulative environmental exposure to the rattiness surface by integrating along a person's trajectory as they move around the community.

Our framework did not account for disease dynamics within rat populations. Given that 80% of rats are estimated to be actively shedding *Leptospira* in Pau da Lima (*Costa et al., 2015b*; *de Faria et al., 2008*) and prevalence in rats is generally high in urban areas globally (*Pellizzaro et al., 2019*; *Costa et al., 2014a*; *Boey et al., 2019*; *Yusof et al., 2019*; *Krøjgaard et al., 2009*), the use of rattiness as a proxy for rat shedding appears reasonable and it may be a useful proxy in other epidemiological studies. Despite this, non-shedding rats may be spatially clustered and future work would benefit from the collection of georeferenced rat infection data. For other zoonotic spillover systems where pathogen release does not occur at a high and homogeneous rate across the reservoir host population, accounting for spatially heterogeneous or time-varying (*Davis et al., 2005*) disease dynamics will be important.

A possible limitation of this study is the titre rise cut-off values used for classifying seroconversion and reinfection in the cohort that determine the sensitivity and specificity of the infection criteria. However, these criteria were used because they are the standard definitions for serological determination of infection that are commonly applied for leptospirosis and a wide range of other infections, and they enable the comparison of results with other previous leptospirosis studies.

The rattiness-infection modelling framework is a flexible tool for exploring the spatial association between reservoir abundance, the environment and human health outcomes. It provides a statistically principled method for joint spatial modelling of infection risk and multiple indices of reservoir abundance, pooling data between indices and directly accounting for uncertainty in their measurement in all parameter estimates and predictions. The framework's geostatistical structure includes spatially continuous predictors for abundance and accounts for spatial correlation, enabling mapping of both infection risk and rattiness. This can be useful for identifying high-risk areas and targeting control. One inherent limitation is its dependence on the availability of spatially continuous environmental variables and abundance data, both of which are prone to high measurement error. This can result in high uncertainty in the model parameter estimates and predictions, as demonstrated by the wider confidence intervals for risk factors in the joint model compared to the standard mixed-effects logistic regression analysis. An additional benefit of the geostatistical structure is that abundance measurements do not have to be taken at the household location, providing some flexibility in the design of eco-epidemiological studies and indices used. The framework may have important applications beyond the study of zoonotic spillover, with the rattiness component replaced by other exposure measures for example mosquito density or ecological indices (such as pollution, where there are multiple, related measures of air or groundwater quality) to model associations with human or animal health outcomes.

In conclusion, we have developed a framework that may have broad applications in delineating complex animal-environment-human interactions during zoonotic spillover and identifying opportunities for public health intervention. We demonstrate its potential by applying it to *Leptospira* in an urban setting, finding evidence that the extent to which local rat shedding drives spillover transmission

is moderated by elevation, most likely a proxy for water runoff. Future work examining these transmission mechanisms in similar settings and across different time points will be key to establishing how generalisable these results are.

## Acknowledgements

We thank the residents and community leaders of Pau da Lima community for their support and participation in this study. This work was supported by the Oswaldo Cruz Foundation and Secretariat of Health Surveillance, Brazilian Ministry of Health, the National Institutes of Health of the United States, the Wellcome Trust and by the Fundação de Amparo à Pesquisa do Estado da Bahia. MTE was supported by a UK Research and Innovation (UKRI) doctorate studentship. FNS was supported by a FAPESB doctorate scholarship.

## Additional information

### Competing interests

Albert I Ko: has received funding from Serimmune and Zoetis for work related to leptospirosis. AIK also received payment and honoraria from Reckit Global Health Institute for participating in a non-profit panel. AIK received travel support from World Health Organisation and Brazilian Ministry of Health. AIK is listed as co-inventor on an issued patent (US 7,718,183 B2) and pending patent (US 61/951,732) related to leptospirosis vaccines. AIK is also on the following boards: Board of Directors, American Society of Tropical Medicine and Hygiene; Executive Board Member (2009-present), International Leptospirosis Society; Member, Inaugural Expert Panel, Reckitt Global Hygiene Institute; Steering Committee Member, Global Leptospirosis Environmental Action Network (GLEAN), WHO. The author has no other competing interests to declare. The other authors declare that no competing interests exist.

### Funding

| Funder | Grant reference number | Author |
|---|---|---|
| National Institutes of Health | F31 AI114245 | Albert I Ko |
| National Institutes of Health | R01 AI052473 | Albert I Ko |
| National Institutes of Health | U01 AI088752 | Albert I Ko |
| National Institutes of Health | R01 TW009504 | Albert I Ko |
| National Institutes of Health | R25 TW009338 | Albert I Ko |
| Medical Research Council | 964635 | Max T Eyre |
| Wellcome Trust | 102330/Z/13/Z | Nivison Nery Federico Costa |
| Fundação Oswaldo Cruz | | Federico Costa |
| Fundação de Amparo à Pesquisa do Estado da Bahia | FAPESB/JCB0020/2016 | Fábio N Souza Federico Costa |

The funders had no role in study design, data collection and interpretation, or the decision to submit the work for publication. For the purpose of Open Access, the authors have applied a CC BY public copyright license to any Author Accepted Manuscript version arising from this submission.

## Author contributions
Max T Eyre, Conceptualization, Data curation, Software, Formal analysis, Investigation, Visualization, Methodology, Writing – original draft, Writing – review and editing; Fábio N Souza, Ticiana SA Carvalho-Pereira, Nivison Nery, Daiana de Oliveira, Jaqueline S Cruz, Gielson A Sacramento, Elsio A Wunder, Kathryn P Hacker, José E Hagan, Data curation, Investigation, Writing – review and editing; Hussein Khalil, James E Childs, Writing – review and editing; Mitermayer G Reis, Conceptualization, Funding acquisition, Investigation, Project administration, Writing – review and editing; Mike Begon, Conceptualization, Supervision, Writing – review and editing; Peter J Diggle, Emanuele Giorgi, Conceptualization, Software, Supervision, Methodology, Writing – review and editing; Albert I Ko, Conceptualization, Funding acquisition, Writing – review and editing; Federico Costa, Conceptualization, Resources, Supervision, Funding acquisition, Investigation, Project administration, Writing – review and editing

## Author ORCIDs
Max T Eyre http://orcid.org/0000-0001-9847-8632
Fábio N Souza http://orcid.org/0000-0002-3542-8918
Ticiana SA Carvalho-Pereira http://orcid.org/0000-0003-2370-2198
Elsio A Wunder http://orcid.org/0000-0002-5239-8511
Albert I Ko http://orcid.org/0000-0001-9023-2339
Emanuele Giorgi http://orcid.org/0000-0003-0640-181X

## Ethics
Human subjects: Participants were enrolled according to written informed consent procedures approved by the Institutional Review Boards of the Oswaldo Cruz Foundation and Brazilian National Commission for Ethics in Research, Brazilian Ministry of Health (CAAE: 01877912.8.0000.0040) and Yale University School of Public Health (HIC 1006006956).

For the rats captured in the rat ecology study, the ethics committee for the use of animals from the Oswaldo Cruz Foundation, Salvador, Brazil, approved the protocols used (protocol number 003/2012), which adhered to the guidelines of the American Society of Mammalogists for the use of wild mammals in research (Sikes and Gannon, 2011) and the guidelines of the American Veterinary Medical Association for the euthanasia of animals (Leary et al., 2013). These protocols were also approved by Yale University's Institutional Animal Care and Use Committee (IACUC), New Haven, Connecticut (protocol number 2012-11498).

## Decision letter and Author response
Decision letter https://doi.org/10.7554/eLife.73120.sa1
Author response https://doi.org/10.7554/eLife.73120.sa2

# Additional files

## Supplementary files
• Transparent reporting form
• Reporting standard 1. STROBE checklist for reporting observational studies.

## Data availability
Rat and human data analysed in this study have been deposited in OSF (https://doi.org/10.17605/OSF.IO/AQZ2Y). However, household coordinates and valley ID have been removed from the human data to ensure participant anonymity. Modelling functions, R scripts and metadata for analyses in this manuscript are publicly available at https://github.com/maxeyre/Rattiness-infection-framework, (copy archived at swh:1:rev:e7953d38269ce97221dbdd83c0be2c65d92dff40).

The following dataset was generated:

| Author(s) | Year | Dataset title | Dataset URL | Database and Identifier |
|---|---|---|---|---|
| Eyre M, Costa F, Ko A | 2021 | Linking rattiness, geography and environmental degradation to spillover Leptospira infections in marginalised urban settings: Data sources | https://doi.org/10.17605/OSF.IO/AQZ2Y | Open Science Framework, 10.17605/OSF.IO/AQZ2Y |

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

## Appendix 1

### Functional form of continuous explanatory variables

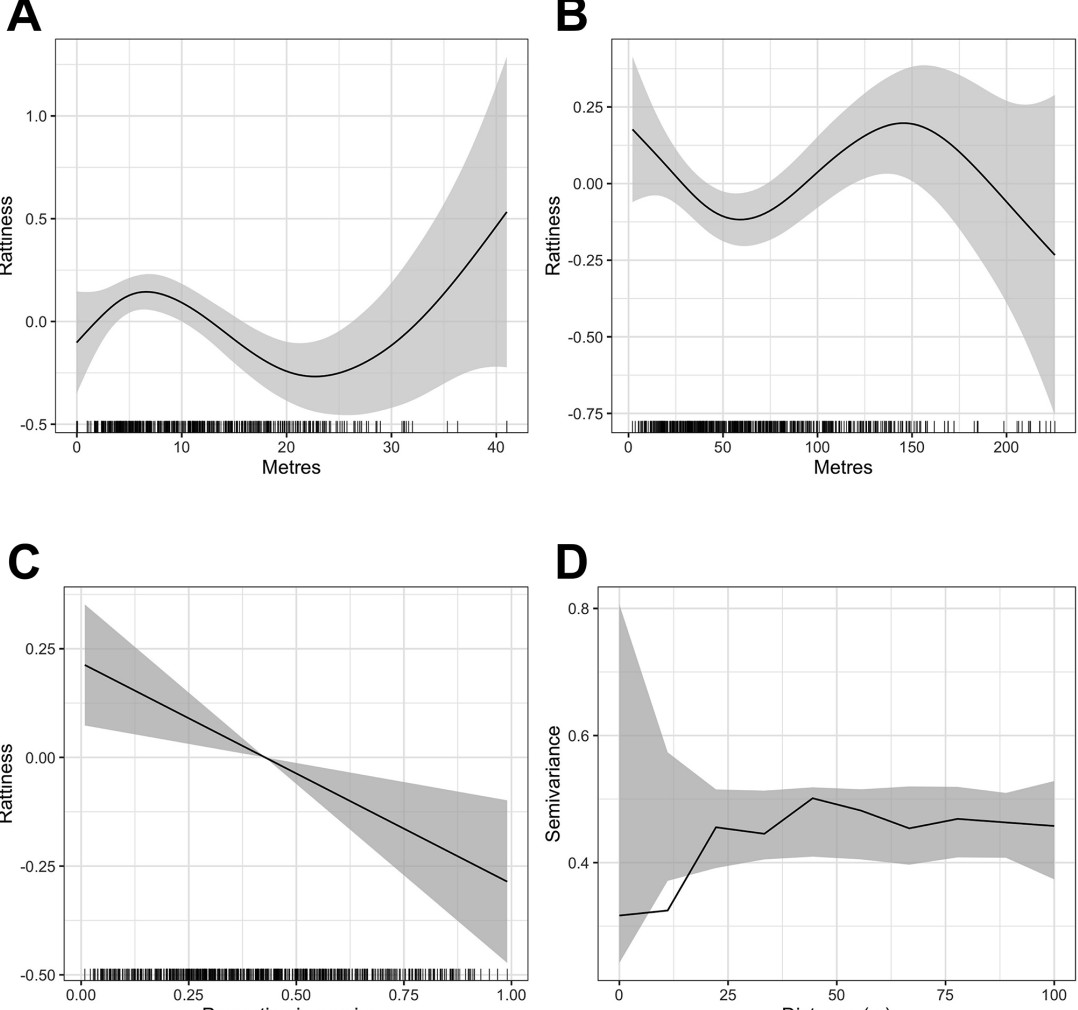

**Appendix 1—figure 1.** Generalized Additive Model (GAM) partial dependence plots for the unstructured random variation in rattiness, $\hat{U}_i$ plotted against the continuous explanatory variables considered in the analysis (shaded areas correspond to 95% confidence intervals). (**A**) elevation relative to the bottom of valley, (**B**) distance to large refuse piles, (**C**) impervious land cover in 20 m radius buffer around sampling point. are estimated using a non-spatial model which excludes all covariates. (**D**) is a variogram computed from $\hat{U}_i$ using a non-spatial model that includes all of the covariates; the dashed lines correspond to 95% confidence intervals under the assumption of spatial independence.

A single knot point for the distance to refuse piles variable was chosen at 50 m to account for the expected decay in the effect of food resources up to a rat home range distance, beyond which little effect would be expected. We did not include an additional knot point at 145 m despite there being a visible change in gradient in *Appendix 1—figure 1* - panel B for two reasons. Firstly, the home range of Norway rats is estimated to be less than 100 m in these urban settings (*Feng and Himsworth, 2014*; *Davis et al., 1948*; *Byers et al., 2019*), meaning that rat abundance is very unlikely to be affected by the availability of anthropogenic food sources beyond this distance, particularly given the high availability of food across the study area. Secondly, we could offer no scientific rationale for why rattiness would start to increase again beyond 50 m before peaking at a very large distance of 145 m from a refuse pile and decreasing thereafter.

In contrast, the mechanisms by which rattiness varies with elevation are more complex, with significant changes in the environment occurring at different elevations. For example, the relationship identified in *Appendix 1—figure 1* - panel A can be explained by the high risk of flooding at the bottom of the valley, which carries resources down to lower elevations (resulting in a peak of rattiness at about 7 m) but makes the very lowest elevations unsuitable for rat burrows. The highest elevations in our study area are close to a main road with food markets where large quantities of food waste are left out in the street for collection. Although there is large uncertainty about this relationship, it is highly possible that this may be driving the positive relationship between 23 m and 40 m.

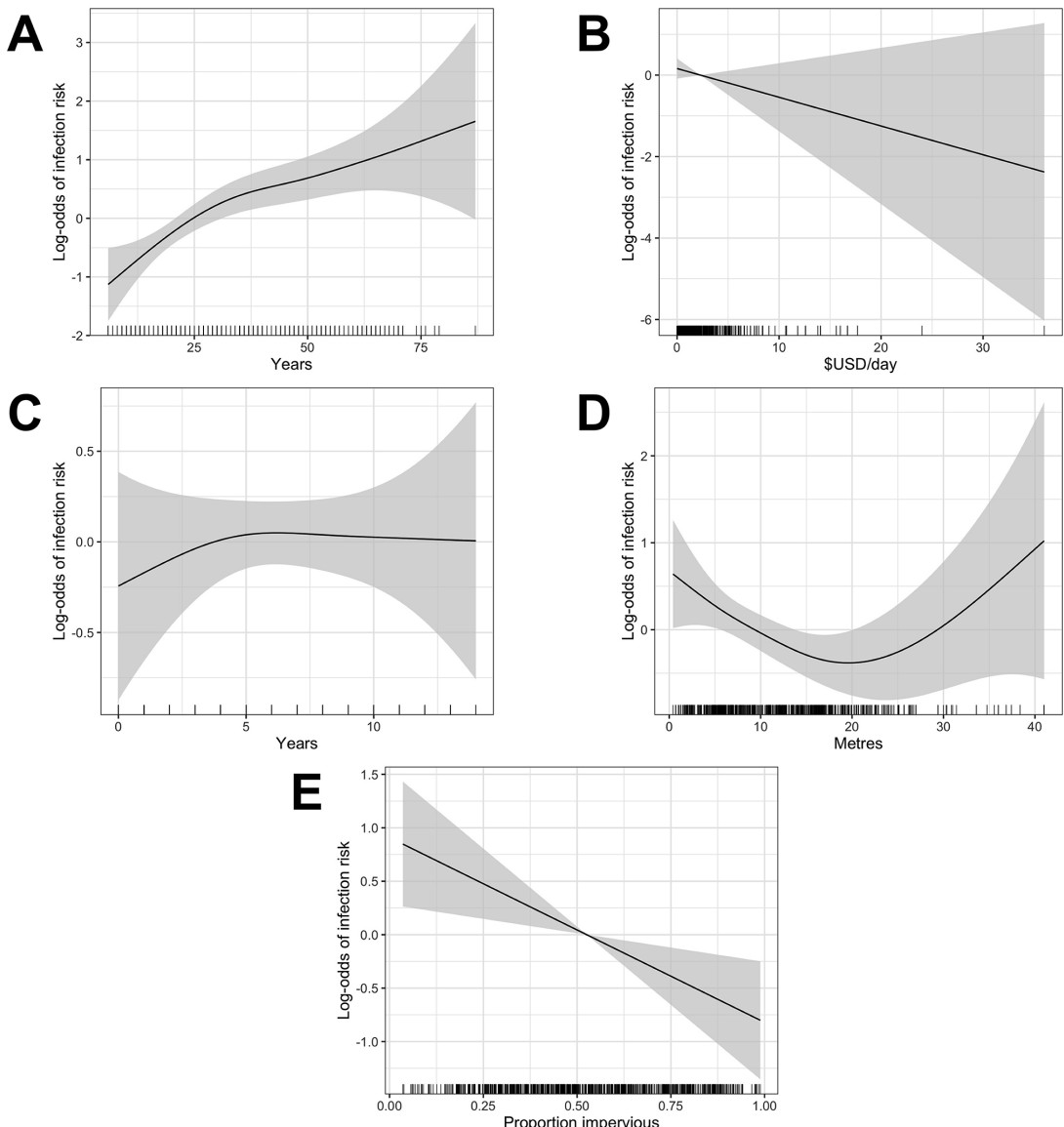

**Appendix 1—figure 2.** Generalized Additive Model (GAM) partial dependence plots for human infection risk plotted against the continuous explanatory variables considered in this analysis (shaded areas correspond to 95% confidence intervals). (**A**) age, (**B**) household per capita income (in USD), (**C**) years of education, (**D**) household elevation relative to the bottom of valley, (**E**) impervious land cover in 20 m radius buffer around household.

Single knot points were considered for age at 30 years (*Appendix 1—figure 2* - panel A), education at 5 years (*Appendix 1—figure 2* - panel C) and relative elevation at 20 metres (*Appendix 1—figure 2* - panel D) based on the value of the explanatory variable at which the gradient of the relationship changed in these plots.

# Appendix 2

## Model selection tables

**Appendix 2—table 1.** AIC fit of the five highest ranked multivariable rattiness models ('+' indicates that a variable was selected in the model).

| Model | Dist. refuse (0–50) | Dist. refuse (>50) | Land cover | Elevation (0–8 m) | Elevation (8–22 m) | Elevation (>22 m) | df* | AICc * |
|---|---|---|---|---|---|---|---|---|
| M1 | + | + | + | + | + | + | 8 | 1476.48 |
| M2 | | | + | + | + | + | 6 | 1479.31 |
| M3 | + | | + | + | + | + | 7 | 1481.01 |
| M4 | + | + | | + | + | + | 7 | 1481.99 |
| M5 | + | + | + | + | | + | 7 | 1482.97 |

*df, degrees of freedom; AICc, corrected Akaike Information Criterion.

**Appendix 2—table 2.** AIC fit of the five highest ranked multivariable human infection models ('+' indicates that a variable was selected in the model).

| Model | Age (0–30) | Age (>30) | Sex | Valley | Floodwater | Income | Land cover | Salesperson | Elevation level | Rattiness | Ratt:Elev | df* | AICc * |
|---|---|---|---|---|---|---|---|---|---|---|---|---|---|
| M1 | + | + | + | + | + | + | | + | + | + | + | 16 | 523.14 |
| M2 | + | + | + | + | + | + | | | + | + | + | 15 | 523.52 |
| M3 | + | + | + | + | + | + | + | + | + | + | + | 17 | 523.72 |
| M4 | + | + | + | + | + | + | + | | + | + | + | 16 | 524.11 |
| M5 | + | + | + | + | | | | + | + | + | + | 14 | 525.04 |
| M*† | + | + | + | + | + | + | | + | + | | | 13 | 532.13 |

*df, degrees of freedom; AICc, corrected Akaike Information Criterion

†Model M* was ranked outside of the top 5 models but is included here for reference to demonstrate the improvement in model fit when rattiness is included.

# Appendix 3

## Exploratory multivariable analysis of human risk factors

**Appendix 3—table 1.** Multivariable mixed effects logistic regression analysis of risk factors for leptospiral infection in community members.

Note: there was missing information for the contact with floodwater question for two individuals and consequently only 1399 participants from 668 households were included in this analysis.

| Variable | OR (95% CI) |
| --- | --- |
| Demographic and social status | |
| Age (per year)* | |
| 0–30 years old | 1.10 (1.04, 1.16) |
| >30 years old | 1.02 (0.96, 1.09) |
| Male gender | 2.90 (1.59, 5.28) |
| Daily per capita household income (US$/day) | 0.93 (0.81, 1.06) |
| Valley | |
| 1 | REF |
| 2 | 3.91 (1.33, 11.68) |
| 3 | 2.26 (0.74, 6.93) |
| Household environment | |
| Relative elevation level | |
| High (>15.6 m) | REF |
| Medium (6.7–15.6 m) | 0.71 (0.30, 1.70) |
| Low (0–6.7 m) | 1.08 (0.44, 2.62) |
| Occupational exposures | |
| Work as travelling salesperson [†] | 3.38 (0.77, 14.87) |
| Behavioural exposures | |
| Contact with floodwater in last 6 months | |
| Never/rarely | REF |
| Sometimes | 0.64 (0.28, 1.43) |
| Frequently | 2.48 (1.02, 6.02) |
| Rattiness | |
| *Rattiness* at high elevation level (per unit *rattiness*) | 6.92 (1.88, 25.47) |
| Elevation level: Low × *rattiness* | 0.10 (0.02, 0.62) |
| Elevation level: Medium × *rattiness* | 0.15 (0.02, 0.91) |
| $\sigma^2$ (variance of household random effect) | 1.78 |

*The effect of age is modelled as a broken linear model with a transition at 30 years old, as informed by the relationship described by Generalized Additive Modelling (***Appendix 1—figure 2***).

[†]Binary variable with reference category of 'no occupational exposure'.

## Appendix 4

### Model fitting

To fit the joint model, we proceed as follows. Let $W = (Y, Z)$ and $\theta = (\alpha_1, ..., \alpha_5, \alpha_h, \sigma_1, ..., \sigma_5, \beta_h, \gamma, \xi, \sigma^2)$ and $\omega = (\beta_r, \phi, \psi)$ be the vector of unknown parameters associated with $[R]$ and $[W|R]$. The likelihood function is then given by

$$L(\theta, \omega) = [W; \theta, \omega] = \int_{\mathbb{R}^N} [R; \omega][W|R; \theta] gtdR \tag{5}$$

The integral in **Equation 5** cannot be solved analytically so we approximate it using Monte Carlo methods. Specifically, let $\theta_0$ and $\omega_0$ be our initial best guesses for $\theta$ and $\omega$, respectively. Since $[R; \omega][W|R; \theta] \propto [R|W; \omega]$ we re-write the integral in **Equation 5** using an importance sampling distribution $[R; \omega_0][W|R; \theta_0]$ to give

$$
\begin{aligned}
L(\theta, \omega) &\propto \int_{\mathbb{R}^N} \frac{[R;\omega][W|R;\theta]}{[R;\omega_0][W|R;\theta_0]} [R|W; \theta_0, \omega_0] \, dR \\
&= E\left[ \frac{[R;\omega][W|R;\theta]}{[R;\omega_0][W|R;\theta_0]} \right],
\end{aligned} \tag{6}
$$

where the expectation is taken with respect to the distribution of $[R|W; \omega_0]$.

Based on **Equation 6**, we then approximate **Equation 5** with

$$L(\theta, \omega) \approx \frac{1}{B} \sum_{b=1}^{B} \frac{[r_{(b)};\omega][W|r_{(b)};\theta]}{[r_{(b)};\omega_0][W|r_{(b)};\theta_0]} \tag{7}$$

where $r_{(b)}$ is the $b$-th sample from $[R|W; \omega_0, \theta_0]$. To obtain the maximum likelihood estimates for $\theta$ and $\omega$, we maximize **Equation 7** using numerical optimization. To simulate from $[R|W; \theta_0, \omega_0]$, we use the Laplace sampling algorithm described in detail by **Christensen, 2004** and **Giorgi and Diggle, 2017**. We draw 110,000 samples from $[R|W; \theta_0, \omega_0]$, with a burn in of 10,000 samples and thin by 10%, leaving 10,000 MCMC samples.

To improve the approximation of the likelihood function, we also update our guesses $\omega_0$ and $\theta_0$ by plugging their estimated values into the denominator of **Equation 7** and iterate its maximization until convergence.

# Appendix 5

## Baseline cohort characteristics

**Appendix 5—table 1.** Summary of demographic, socioeconomic and environmental risk factors.

| Variable | No. or Median (% or IQR) * |
|---|---|
| **Demographic and social status** | |
| Age (years) | 27 (15–41) |
| Male gender | 597 (42.6%) |
| Daily per capita household income (US$/day) | 1.6 (0.8–2.8) |
| Valley 1 | 259 (18.5%) |
| Valley 2 | 557 (39.8%) |
| Valley 3 | 585 (41.8%) |
| Literacy | 1125 (80.3%) |
| Education (years) | 6 (4-9) |
| **Household environment** | |
| Impervious land cover (%) | 49.6 (35.1–70.6) |
| Relative elevation (metres) | 11.0 (5.9–16.3) |
| Elevation level | |
| Low (0–6.7 m) | 474 (33.8%) |
| Medium (6.7–15.6 m) | 524 (37.4%) |
| High (>15.6 m) | 403 (28.8%) |
| Open sewer within 10 m | 926 (66.1%) |
| Unprotected from open sewer | 666 (47.6%) |
| Live on hillside | 453 (32.4%) |
| **Occupational exposures** | |
| Work in construction | 105 (7.5%) |
| Work as travelling salesperson | 24 (1.7%) |
| Work in refuse collection | 61 (4.4%) |
| Work involves contact with mud | 27 (1.9%) |
| Work involves contact with floodwater | 23 (1.6%) |
| Work involves contact with sewer water | 16 (1.1%) |
| **Behavioural exposures** | |
| Contact with floodwater in last 6 months | |
| Never/rarely | 986 (70.5%) |
| Sometimes | 299 (21.4%) |
| Frequently | 114 (8.1%) |
| Contact with sewer water in last 6 months | |
| Never/rarely | 1120 (80.2%) |
| Sometimes | 180 (12.9%) |
| Frequently | 97 (6.9%) |

* No., number; IQR, interquartile range; Percentages are calculated without missing values. All variables had ≤ 5 missing values.

## Appendix 6

### Sensitivity analysis for disturbed trap modelling assumption

In the rattiness-infection framework we assumed that a trap was disturbed when it was found closed without a rat and set $t = 0.5$ (see 'Rat abundance outcomes') in the equation for the probability of capturing a rat

$$1 - \exp\{-t_i \mu_1(x_i)\}.$$

This occurred in 554 (36.6%) out of 1,512 trapping-days. To ascertain the potential impact of this on model parameter estimates we conducted a sensitivity analysis as follows:

1. Draw values for t from $U(0, 1)$ for all trap observations that were found closed.
2. Fit a simplified rattiness model with covariates that did not account for spatial correlation by setting $\psi = 0$ in *Equation 2* in 'Rattiness'.
3. Repeat steps 1–2 a total of 1,000 times.
4. Estimate the between-imputation standard error for each parameter, defined as:

$$SE_{imp} = \sqrt{\frac{\sum_{i=1}^{B}(\theta_i - \bar{\theta}^2)}{B - 1}}$$

for imputation $i$ of a total $B$ imputed datasets.

The results for each parameter can be seen in *Appendix 6—table 1* below. Estimated between-imputation standard errors were small relative to parameter estimates, indicating that uncertainty due to the missing trap disturbance information is unlikely to have significantly affected parameter estimates in the full rattiness-infection model.

**Appendix 6—table 1.** Trap disturbance sensitivity analysis: non-spatial rattiness model parameter estimates and between-imputation standard errors.

| Parameter | Estimate | $SE_{imp}$ |
|---|---|---|
| $\alpha_{traps}$ | –2.8274 | 0.0128 |
| $\alpha_{plates}$ | –1.9058 | 0.0004 |
| $\alpha_{burrows}$ | –1.3794 | 0.0008 |
| $\alpha_{faeces}$ | –2.8617 | 0.0027 |
| $\alpha_{trails}$ | –2.1538 | 0.0023 |
| $\sigma_{traps}$ | 0.7010 | 0.0120 |
| $\sigma_{plates}$ | 2.4016 | 0.0004 |
| $\sigma_{burrows}$ | 1.3820 | 0.0008 |
| $\sigma_{faeces}$ | 2.6704 | 0.0031 |
| $\sigma_{trails}$ | 2.6431 | 0.0036 |
| Relative elevation (per 1 m increase)[2] | | |
| 0–8 m | 0.0525 | 0.0001 |
| 8–22 m | –0.0583 | 0.0001 |
| >22 m | 0.1112 | 0.0002 |
| Distance to large refuse piles (per 10 m increase)[3] | | |
| 0–50 m | –0.1090 | 0.0002 |
| >50 m | 0.0405 | 0.0001 |
| Impervious land cover (per 10% increase) | –0.0592 | 0.0001 |

## Appendix 7

### Residual diagnostics

To examine the fit of the full rattiness-infection model to the human infection data, a formal diagnostic investigation was conducted using randomized quantile residuals (*Dunn and Smyth, 1996*; *Smyth et al., 2021*). The residual plots in *Appendix 7—figure 1* exhibit no trends between quantile residuals and fitted values (Panel A) or variables in the model (Panels B-H).

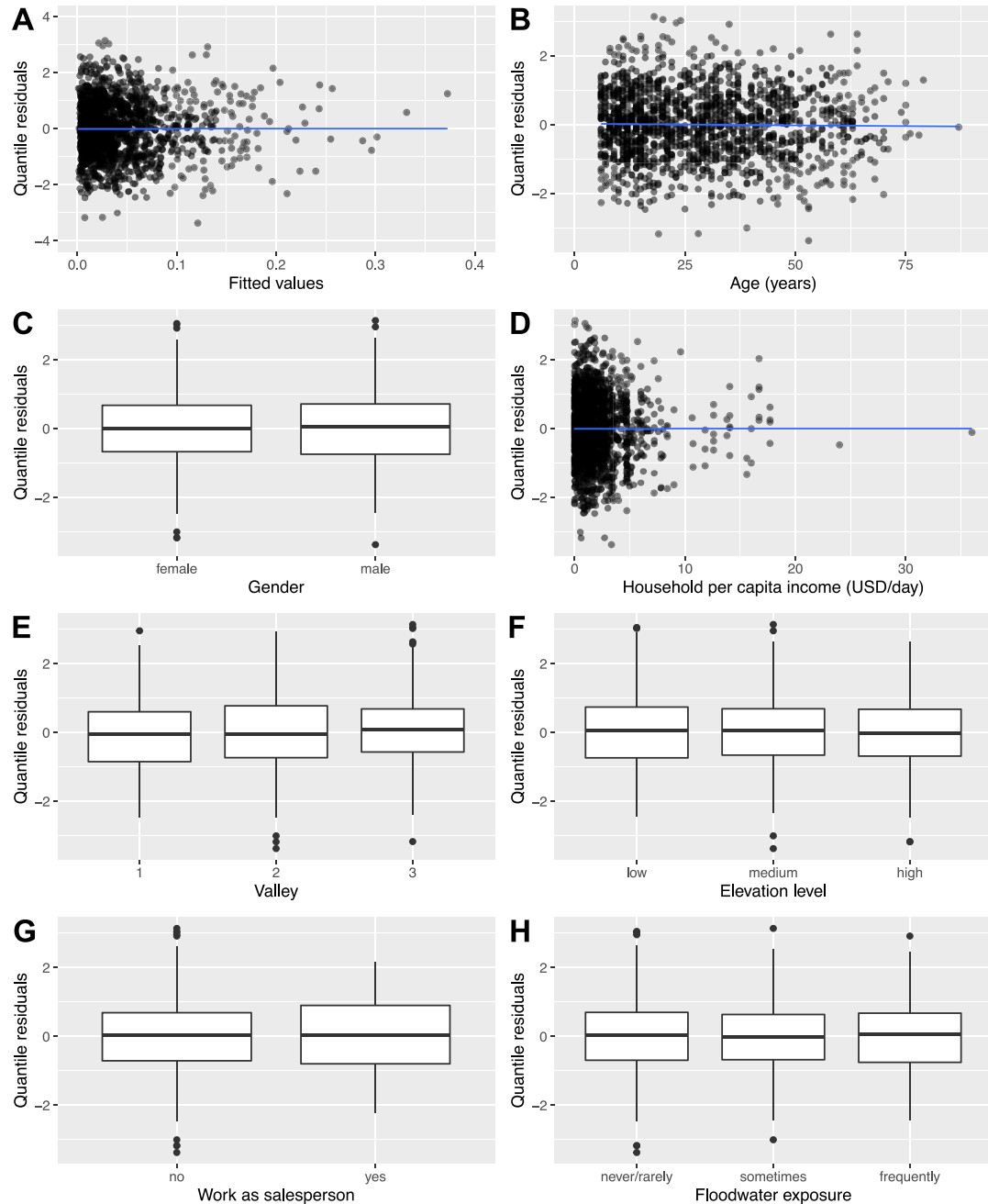

**Appendix 7—figure 1.** Residual diagnostic plots showing randomised quantile residuals plotted against: (**A**) fitted values; (**B–H**) variables in the model.

## Appendix 8

## Model building guidance

To help guide the model building process for future users of the rattiness-infection framework we outline the following key steps (to be viewed with the available R code at https://github.com/maxeyre/Rattiness-infection-framework):

1. Set up the rat (or any other animal reservoir) component of the model [script: 1-rat-explore.R]:
    a. Fit the non-spatial rattiness model with no explanatory variables and predict rattiness at all sampled rat locations.
    b. Explore the relationship between predicted mean rattiness and explanatory variables using Generalized Additive Models (GAMs) to decide on their functional form. Note: all variables considered must also be measured at household locations
    c. Conduct model selection using a linear model with mean predicted rattiness as the dependent variable.
    d. Fit the non-spatial rattiness model with selected explanatory variables and compute the empirical variogram to check for evidence of residual spatial autocorrelation. If the variogram shows no signs of residual correlation, consider confirming this result by fitting the model in the following step - the estimated scale of spatial correlation $\phi$ should be close to zero (it may also not be able to estimate its value, with the value changing considerably between iterations).
    e. Fit the spatial rattiness model with selected explanatory variables and predict mean rattiness at household locations to create an exploratory rattiness variable. To improve model convergence use parameter estimates from the non-spatial model as the first guess for the parameters and repeat model fitting by plugging in previous estimates.
2. Set up the human infection component of the model [script: 2-human-explore.R]:
    a. Explore the relationship between infection risk and explanatory variables using Generalized Additive Models (GAMs) to decide on their functional form.
    b. Conduct model selection
    c. Explore the relationship between infection risk and mean predicted rattiness using Generalized Additive Models (GAMs) and consider interactions where relevant.
    d. Test for residual spatial correlation after controlling for selected variables and mean predicted rattiness (we recommend using the PrevMap package).
3. Using the joint rattiness-infection framework:
    a. Fit the joint model [script: 3-fit-joint-model.R]. In the 'control' file, the inclusion of a household-level random effect (or 'nugget') and an additional spatial Gaussian process (if there was evidence of residual spatial correlation after controlling for explanatory variables and rattiness in the previous step) in the human linear predictor can be controlled. We recommend monitoring parameter estimates from each iteration to assess how well the model is converging. If parameters for the spatial Gaussian processes are not converging then this may indicate that the data do not support the inclusion of an additional spatial Gaussian process in the human component and a simpler model should be considered. This model fitting process can be time consuming and ideally should be run on a high-end computing network.
    b. Conduct a residual diagnostic analysis for the full rattiness-infection model [script: 5-revision subanalyses.R].
    c. Bootstrap to estimate uncertainty in parameter estimates [script: 3-fit-joint-model.R]
    d. Create prediction maps for rattiness, infection risk and spatial Gaussian processes (if required) [script: 4-spatial-prediction.R]. The prediction grid for your study area must include values for all variables included in the model.

