## [Editor Report]

In their work, the authors present a novel geostatistical framework allowing for modelling complex animal-environment-human interactions during zoonotic spillover. The presented case relates to zoonotic spillover of Leptospira infections in a marginalised urban setting in Salvador, Brazil. The outcomes of such applications could contribute to inform public health interventions. The methodological approach is to be applauded and can be of benefit beyond the study of zoonotic spillover.

---

## [Decision Letter]

**Decision letter after peer review:**

Thank you for submitting your article "Linking rattiness, geography and environmental degradation to spillover *Leptospira* infections in marginalised urban settings: an eco-epidemiological community-based cohort study in Brazil" for consideration by *eLife*. Your article has been reviewed by 3 peer reviewers, one of whom is a member of our Board of Reviewing Editors, and the evaluation has been overseen by David Serwadda as the Senior Editor. The following individual involved in review of your submission has agreed to reveal their identity: Benny Borremans (Reviewer #3).

Essential revisions:

The reviewers and I have several comments that should be carefully addressed. I tried to merge the essential ones here; though the public reviews and the recommendations for authors should be taken into account to (note that there is some overlap between the essential revisions and the public reviews).

1) On the statistical model and the choices made:

– On p7, section 2.2.2 the authors use mu_2 for defining the intensity of the inhomogeneous Poisson process. Shouldn't this be mu_1 rather mu_2? If not, what makes using the same function for Yi,1 and Yi,2 a reasonable choice? Note that the model used in both components is not the same.

– Did the authors perform a sensitivity analysis related to the assumption of t=0.5 for the disturbed traps (how often did this occur)?

– For the most part, the explanatory variables assessed in the different models were well described and justified, however there were some cases for which further explanation would have been helpful. For example, how did the authors determine which occupations to evaluate? Specifically, why traveling salesperson? What is the difference between open sewer within 10 m and unprotected from sewer?

– Sup file 2 and Figure S1 (and Table 1): This could be a function of me not understanding correctly, not necessarily the authors not conducting the study appropriately, but I couldn't understand why elevation was split into 3 when distance to large refuse piles was only split into 2 categories since the shape of the splines was similar. Based on Figure S1 (B) it seems that the effect decreases until ~ 60m then increases until ~ 145m and then decreases again? Also, it was unclear to me why in Table 1 an effect estimate for distance to large refuse piles of.02 is 'of little effect' when one of -.07 is considered noteworthy. They both seemed quite small.

– Table 2: It was unclear to me why both relative elevation and elevation level were included and how they differed. Further explanation would be helpful.

– Figure 4. It seems to me that the elevation levels were chosen simply by identifying the elevation cut-offs that divided the household sample sizes into three equal groups. It would be helpful if the authors included a viable biological justification for this division.

– The authors provide an extensive model building exercise and investigate, in different ways, whether the model captures the necessary complexity (GAM smoothers – testing linearity, spatial correlation, etc). I believe the work would benefit from (1) a formal diagnostic investigation, if feasible; (2) providing guidelines on how model building should be performed.

More specifically there are some additional concerns about this specific analysis:

(1) The infection risk data: while the actual infection risk data are not shown, the map shown in Figure 5B suggests that there is an infection hotspot that happens to be at high elevation. This raises the question of how strongly this single hotspot is driving the observed correlation between rat abundance and infection risk (which the authors find to be much stronger at high elevation than at lower elevations).

(2) The statistical models: if I understand correctly, all tested models of infection risk include the variable rat abundance, and while the individual effect estimates for rat abundance are statistically significant (Table 3), the more important question of how the fit of a model without the rat abundance variables compares with those of the other tested models (shown in Supplementary Table S2) has not been addressed.

I am wondering about this curious spatial pattern, where there seems to be one main predicted hotspot of infection risk (Figure 5B), which happens to be at a high elevation. There are a few other locations at a similar elevation, but these don't result in high infection risk predictions, which I assume is because of a difference in other important covariates? When comparing this result to the rattiness map (Figure 5A), one would never guess there is a meaningful (biologically significant) correlation between rattiness and infection risk. Model selection however did find a statistically significant effect of rattiness (Table 3), with the largest effect sizes for the high elevation. This makes me wonder whether the statistical pattern is mostly driven by this one hotspot that happens to be at high elevation, and how important rattiness really is overall.

It would be great to see a map based on the raw infection data, so it's possible to get a better sense of this possible biasing effect on the contribution of rattiness. Maybe add it to figure 5?

One way to test this would be to do the same analysis, but without the location(s) driving this high infection risk hotspot, and see if rattiness is still an important contributor to infection risk.

Perhaps more importantly, all human infection models (Supplementary table 2) include rattiness, so there is no way to assess how a model without rattiness compares with those that do. I strongly suggest adding at least one model without rattiness, for example model M1 but excluding rattiness. If the AIC values of all models in Table S2 are much lower than a model without rattiness, it would add a lot of confidence to the assumed significant effect of rattiness. This is related to the model framework relying on conditional independence within its built up (equation (1)). Whereas this is a reasonable assumption, it would be good to discuss situations in which this assumption is questionable and what the implications are for applying the modeling framework to other settings. In addition the authors indicate that the most complex model was chosen when modeling rattiness (p8, section 2.3.1). Doesn't this imply that the model selection reaches its limits given the candidate models at hand, ie is there a need to consider more complex models?

2) Presentation and interpretation of results

– In Tables 2 and 3, the authors present their results in a comprehensive way but it's not easy to connect those tables in terms of results. For example; are the occupational exposures binary variables? If not, what is the reference category and why is only one (work as traveling salesperson) retained in Table 3? Which of the variables reach overall significance?

*Reviewer #1 (Recommendations for the authors):*

I believe the manuscript is overall clearly written. I do have a few questions for clarification though.

On p2, section 2.1.2 serosurveys, eligibility criteria for inclusion in the cohort study are outlined. It would help explaining why these specific conditions were used for inclusion: ie 'who had slept more than 3 nights in the previous week in a study household'.

On p7, section 2.2.3 the authors define Zi,j using a Bernoulli variable with probability p_j(x_i). Wouldn't it make sense to consider a hazard-based framework or derive the corresponding hazard function in terms of it's interpretation?

Textual comments:

– Please use \mbox for the correlation in section 2.2.1.

*Reviewer #2 (Recommendations for the authors):*

Line 62 – typo? Analyze vs. analysing?

Figure 2 description – typo? … dh and dr are not mutually exclusive groups of explanatory [variables – missing?] and the same variables…

Line 277 – it would be nice to reference table 2 here so that the readers can see the full list of considered variables by group.

Regarding supplemental information, it would have been easier if the actual table or figure had been referenced vs. the file in which it could be found.

*Reviewer #3 (Recommendations for the authors):*

It was a pleasure to review your manuscript.

In my opinion the writing is excellent, the study design is clever and powerful (and must have been a lot of work!), and the spatial statistics are performed expertly.

I do have a few suggestions, that I hope can either be easily refuted or can help to improve the analyses.

Congratulations on this fantastic work.

L65: I suggest writing DALYs in full, as not all readers will know what this is.

L94: I don't agree that there is an absence of methods (multilevel Bayesian models for example have been around for a while), but rather that they are rarely applied in this context.

L99: I find this particular unspecific use of abundance quite confusing, as this is already a very specific and well-defined ecological term. For example, what exactly is then meant by reservoir host abundance on L104? Is this the number of reservoir hosts, or the number of infected hosts, or the number of leptospires in the environment?

If it is used as a measure of exposure to a disease of interest (L100), why not use a term like pathogen pressure, or just exposure? I strongly suggest using different words to describe actual abundance and pathogen-related abundance.

L115: The term rattiness is useful (and fun), but does it really represent leptospire pressure by rats if the model does not take into account leptospira prevalence/shedding in the rat populations? I agree that the presence/abundance of rats can be a decent proxy for the potential risk of leptospira spillover in locations with known presence of leptospira in the rat population, but I'm less inclined to accept that it is ok to define rattiness, which implies rat abundance, as a proxy for leptospiral contamination when the study did not measure the presence of leptospira in the rats.

I see that this is mentioned in the discussion (L493). I think it might be more useful to add this information earlier on, at the place where the term rattiness is introduced.

I agree that with such a high prevalence, it is reasonable to use rattiness as a proxy, but would still be wary: these 80% of rats are likely not distributed randomly across the area, as pathogen transmission is typically more spatially clustered. That means that 1 out of 5 local rat populations are not infected, which is definitely not negligible. That is an important caveat to highlight clearly, early on.

L208, 213: On L208, i is defined as a location, and on L213 as household location. I assume these represent the same location? If so, it might be best to be a bit more specific in the definition on L208, and add 'household', just so it's clear there is only one definition of a location.

L284: What is the rationale for choosing those specific knots?

L324: Kudos for citing the individual R packages (as one should, but often not done).

---

## [Author Response]

Essential revisions:1) On the statistical model and the choices made:– On p7, section 2.2.2 the authors use mu_2 for defining the intensity of the inhomogeneous Poisson process. Shouldn't this be mu_1 rather mu_2? If not, what makes using the same function for Yi,1 and Yi,2 a reasonable choice? Note that the model used in both components is not the same.

Thank you for pointing out this typographical error, it has been corrected to mu_1.

– Did the authors perform a sensitivity analysis related to the assumption of t=0.5 for the disturbed traps (how often did this occur)?

We have included the results of a sensitivity analysis to evaluate the impact of this assumption on our model parameter estimates in Appendix 6. In brief, for this analysis we drew independent samples for t from the uniform distribution for all 554 trap observations that closed early (36.6% of n=1,512 trap observations) and fitted a simplified rattiness model to the simulated data. This was repeated 1,000 times and between-imputation standard errors were computed. Estimated between-imputation standard errors were very small relative to parameter estimates (the maximum value of a SE divided by the point estimate was equal to 1.7%, with the rest below 1%), indicating that uncertainty due to the missing trap disturbance information is unlikely to have materially affected estimates of uncertainty in the full rattiness-infection model. This evidence supports the use of the t=0.5 assumption in our full analysis. We have added the following text to Section 2.2.2:

“We conducted a sensitivity analysis for this assumption (see Appendix 6) and found that it did not materially affect rattiness parameter estimates.”

– For the most part, the explanatory variables assessed in the different models were well described and justified, however there were some cases for which further explanation would have been helpful. For example, how did the authors determine which occupations to evaluate? Specifically, why traveling salesperson? What is the difference between open sewer within 10 m and unprotected from sewer?

We have added the following additional text to Section 2.3.2 on line 297 to clarify the definition and reason for inclusion for these variables:

“In the household environment domain, two variables were used to capture risk due to sewer flooding close to the household: (i) the presence of an open sewer within 10 metres of the household location and (ii) a binary `unprotected from open sewer' variable which identified those households within 10 metres of an open sewer that did not have any physical barriers erected to prevent water overflow. Three high-risk occupations were included in the occupational exposures domain as binary variables. Construction workers and refuse collectors have direct contact with potentially contaminated soil, building materials and refuse in areas that provide harbourage and food for rats. Travelling salespeople have regular and high levels of exposure to the environment (particularly during flooding events) as they move from house to house by foot. Two other binary occupational exposure variables were included that measured whether a participant worked in an occupation that involves contact with floodwater or sewer water.”

– Sup file 2 and Figure S1 (and Table 1): This could be a function of me not understanding correctly, not necessarily the authors not conducting the study appropriately, but I couldn't understand why elevation was split into 3 when distance to large refuse piles was only split into 2 categories since the shape of the splines was similar. Based on Figure S1 (B) it seems that the effect decreases until ~ 60m then increases until ~ 145m and then decreases again?

We have added the following text in Appendix 1 to explain the decision to model distance to refuse piles in this way and clarify the difference for elevation:

“A single knot point for the distance to refuse piles variable was chosen at 50m to account for the expected decay in the effect of food resources up to a rat home range distance, beyond which little effect would be expected. We did not include an additional knot point at 145m despite there being a visible change in gradient in Figure S1B for two reasons. Firstly, the home range of Norway rats is estimated to be less than 100m in these urban settings [1-3], meaning that rat abundance is very unlikely to be affected by the availability of anthropogenic food sources beyond this distance, particularly given the high availability of food across the study area. Secondly, we could offer no scientific rationale for why rattiness would start to increase again beyond 50m before peaking at a very large distance of 145m from a refuse pile and decreasing thereafter.

In contrast, the mechanisms by which rattiness varies with elevation are more complex, with significant changes in the environment occurring at different elevations. For example, the relationship identified in Figure S1A can be explained by the high risk of flooding at the bottom of the valley, which carries resources down to lower elevations (resulting in a peak of rattiness at about 7m) but makes the very lowest elevations unsuitable for rat burrows. The highest elevations in our study area are close to a main road with food markets where large quantities of food waste are left out in the street for collection. Although there is large uncertainty about this relationship, it is highly possible that this may be driving the positive relationship between 23m and 40m.”

Also, it was unclear to me why in Table 1 an effect estimate for distance to large refuse piles of.02 is 'of little effect' when one of -.07 is considered noteworthy. They both seemed quite small.

Thank you for pointing this out. We have added the following additional text to Section 3.2 to clarify this: “there was a smaller increase in rattiness of 0.02 (95\%CI -0.05, 0.09) per 10m.”

– Table 2: It was unclear to me why both relative elevation and elevation level were included and how they differed. Further explanation would be helpful.

We have changed ‘relative elevation level’ to ‘relative elevation category’ in the text to distinguish more clearly between these two parameterisations. These are two different parameterisations for household elevation above the valley floor: (i) relative elevation – a continuous variable modelled as a linear spline with a knot at 20 metres; (ii) relative elevation category – a categorical variable modelled as a piecewise constant function with breaks at 6.7 and 15.6 metres resulting in three categories: low, medium and high elevation levels. We have added the following additional text in Section 2.2.4 to clarify this:

“a categorical parameterisation of household elevation relative to the bottom of the valley (modelled as a piecewise constant function with breaks at 6.7 and 15.6 metres, resulting in three categories: low, medium and high elevation levels.)”. We have also added the following footnote to Table 2: “Relative elevation category consists of three discrete groups representing three regions with different flooding risk profiles.”.

Both of these variables were included in the model selection because relative elevation has conventionally been modelled as a continuous variable in earlier studies and we wished to maintain consistency. Relative elevation category was included because our primary aim was to test the hypothesis of whether the role of rats in driving transmission varied across the chosen three elevation categories (low, medium and high) due to their different flood risk profiles. We have added additional text to clarify the reason for the inclusion of the relative elevation category variable.

– Figure 4. It seems to me that the elevation levels were chosen simply by identifying the elevation cut-offs that divided the household sample sizes into three equal groups. It would be helpful if the authors included a viable biological justification for this division.

The choice of these groupings was based on our observations on the spatial variation in flooding risk and leptospirosis risk at low, medium and high elevations in the community as defined in earlier studies conducted at this site. Consequently, our study area was defined to have a roughly even number of households across this elevation gradient. We have added the following additional text to Section 2.2.4 to clarify this:

“This was implemented by first dividing the study area into three elevation categories with different flooding risk profiles (as observed during our work in the study area over the last 15 years): low (0-6.7m from bottom of valley; high flooding risk with maintenance of floodwater for long periods), medium (6.7-15.6m; moderate flooding with high water runoff) and high (>15.6m; limited flooding and water runoff). Our study was then designed to evenly sample across this elevation gradient and minimum and maximum values for each elevation category were chosen to include an equal number of households in each level.”

– The authors provide an extensive model building exercise and investigate, in different ways, whether the model captures the necessary complexity (GAM smoothers – testing linearity, spatial correlation, etc). I believe the work would benefit from (1) a formal diagnostic investigation, if feasible;

We have added a new Appendix 7 with diagnostic plots of randomized quantile residuals to check the rattiness-infection model fit with the human infection data and included the following text in Section 2.4 of the main text:

“A formal diagnostic investigation of randomized quantile residuals is included in Appendix 7. We found no evidence in the diagnostic plots to suggest that there were issues with our modelling approach.”

(2) providing guidelines on how model building should be performed.

To supplement the R code that is publicly available for repeating all of the steps in this analysis, we have now also included a detailed step-by-step explanation of the model building process in Appendix 8 that outlines the key steps for building the rat and infection components of the model (variable selection and evaluation of residual spatial autocorrelation) and fitting and examining the joint rattiness-infection model. We have added the following text in Section 2.6 of the main text:

“We also include a step-by-step explanation of the model building process to guide future users of the rattiness-infection framework in Appendix 8.”

More specifically there are some additional concerns about this specific analysis:(1) The infection risk data: while the actual infection risk data are not shown, the map shown in Figure 5B suggests that there is an infection hotspot that happens to be at high elevation. This raises the question of how strongly this single hotspot is driving the observed correlation between rat abundance and infection risk (which the authors find to be much stronger at high elevation than at lower elevations).

Please see below.

(2) The statistical models: if I understand correctly, all tested models of infection risk include the variable rat abundance, and while the individual effect estimates for rat abundance are statistically significant (Table 3), the more important question of how the fit of a model without the rat abundance variables compares with those of the other tested models (shown in Supplementary Table S2) has not been addressed.I am wondering about this curious spatial pattern, where there seems to be one main predicted hotspot of infection risk (Figure 5B), which happens to be at a high elevation. There are a few other locations at a similar elevation, but these don't result in high infection risk predictions, which I assume is because of a difference in other important covariates? When comparing this result to the rattiness map (Figure 5A), one would never guess there is a meaningful (biologically significant) correlation between rattiness and infection risk. Model selection however did find a statistically significant effect of rattiness (Table 3), with the largest effect sizes for the high elevation.This makes me wonder whether the statistical pattern is mostly driven by this one hotspot that happens to be at high elevation, and how important rattiness really is overall.It would be great to see a map based on the raw infection data, so it's possible to get a better sense of this possible biasing effect on the contribution of rattiness. Maybe add it to figure 5?One way to test this would be to do the same analysis, but without the location(s) driving this high infection risk hotspot, and see if rattiness is still an important contributor to infection risk.

Thank you for this comment. We have added a new figure (Figure 4) earlier on in the article (we decided to add this here rather than to Figure 6 – formerly Figure 5 – to ensure that the map is large enough that points in Figure 4A are easily visible – please note that it is included as a larger and easier to view image in the main *eLife* template version) with the raw infection data overlaid on contour lines for the three elevation levels to provide the reader with a better overview of the raw data. This new Figure 4 shows that out of a total of 403 participants in the high elevation region there were 16 infections, of which only 5 (31%) were located in the large hotspot in Valley 3 (valleys are numbered 1 to 3 from west to east, see Figure 1A). In addition to the largest hotspot in the north of Valley 3, there are several other areas in the high elevation region with raised predicted infection risk values relative to their surroundings where there were also rattiness hotspots and infected participants in the raw data: fives cases (red and yellow infection risk areas in Figure 5B) on the western side of Valley 2; the two cases on the eastern edge of Valley 2; the two cases on the western edge of Valley 3; and the single case in the southwest of Valley 3. Other variables are also important drivers of infection risk and at several of these locations the contribution of rattiness increases infection risk significantly relative to the low-risk surrounding area (e.g. to 10% in areas where risk is closer to 1% or 2%) without reaching the more obviously visible high infection risk values closer to 20%. We believe that our statistical model provides a better test of whether there is a statistical association between rattiness and infection at high elevations than a visual examination, but that this is supported by the large number of observations in the high elevation area (403) and the distribution of infected and uninfected households, which demonstrates that the observed association is not only driven by the hotspot in Valley 2.

Perhaps more importantly, all human infection models (Supplementary table 2) include rattiness, so there is no way to assess how a model without rattiness compares with those that do. I strongly suggest adding at least one model without rattiness, for example model M1 but excluding rattiness. If the AIC values of all models in Table S2 are much lower than a model without rattiness, it would add a lot of confidence to the assumed significant effect of rattiness.

Thank you for pointing this out. These models were considered but were ranked outside of the top five models and for this reason were not reported in Table S2. We agree that showing the AIC of a model without rattiness in this table can more clearly demonstrate the improved fit of the model with rattiness. To do this we have added the highest ranked model without rattiness (M*) to Table S2 and added a note to the table explaining the reason for its inclusion (“Model M* was ranked outside of the top 5 models but is included here for reference to demonstrate the improvement in model fit when rattiness is included”). The AIC of M* was 532.13. This is substantially higher than the top five models (M1 = 523.14 and M5 = 525.04), justifying its inclusion in this model and in the joint rattiness-infection framework.

This is related to the model framework relying on conditional independence within its built up (equation (1)). Whereas this is a reasonable assumption, it would be good to discuss situations in which this assumption is questionable and what the implications are for applying the modeling framework to other settings.

We have added the following text immediately after “is shown schematically in Figure 2” following equation (1) on line 225:

“The conditional independence assumption in (1) is reasonable for a vector-borne disease or one that is transmitted indirectly, in which context the observed rat indices are to be considered as noisy indicators of the unobservable spatial variation in the extent to which the environment is contaminated with rat-derived pathogen. It would be more questionable for applications in which the disease of interest is spread by direct transmission from rat to human.”

In addition the authors indicate that the most complex model was chosen when modeling rattiness (p8, section 2.3.1). Doesn't this imply that the model selection reaches its limits given the candidate models at hand, ie is there a need to consider more complex models?

The number of variables available for consideration in this rattiness model was limited to the three variables included in the model because of the requirement that they were also measured at all household locations. For this reason, we were unable to consider any other environmental variables. In terms of the functional forms of the included variables, we wished to maintain interpretability in the modelled relationships and decided to model them with linear splines rather than considering more complex and less interpretable smoothing functions.

2) Presentation and interpretation of results– In Tables 2 and 3, the authors present their results in a comprehensive way but it's not easy to connect those tables in terms of results. For example; are the occupational exposures binary variables? If not, what is the reference category and why is only one (work as traveling salesperson) retained in Table 3? Which of the variables reach overall significance?

These occupational exposure variables are binary. We have now added additional text in Section 2.3.2 on lines 294 and 297 to clarify this and have added a footnote explaining that the reference category was participants who do not have this exposure in Tables 2 and 3: “Binary variable with reference category of `no occupational exposure'.”. A reference category was not included for each variable because Tables 2 and 3 are already large and we wished to keep it on a single page.

In terms of the overall significance, we prefer to show on confidence intervals to focus on range of possible effect sizes compatible with the data rather than p-values. However, all OR confidence intervals that did not include the value one are significant at the conventional 5% level. To make this more consistent in the text we have edited the text as follows:

Line 394 – “Variables in two of the four domains (demographic and social status and behavioural exposures) had estimated effect sizes with 95\% confidence intervals that did not include an odds ratio of one (statistically significant at the conventional 5\% level).”

Line 422 – “Individuals living in the medium (OR 0.77 95\%CI 0.31, 1.66) and high (OR 0.67 95\%CI 0.11, 1.64) elevation areas had a lower estimated odds of infection relative to those living in the low relative elevation category area where there are open sewers and flooding risk is higher, however these confidence intervals included an odds ratio of one (not statistically significant at the conventional 5% level).”

Reviewer #1 (Recommendations for the authors):I believe the manuscript is overall clearly written. I do have a few questions for clarification though.On p2, section 2.1.2 serosurveys, eligibility criteria for inclusion in the cohort study are outlined. It would help explaining why these specific conditions were used for inclusion: ie 'who had slept more than 3 nights in the previous week in a study household'.

We have added the following additional text to clarify this “This study focussed on ground floor households because they are vulnerable to flooding and consequently at high risk for leptospiral transmission. The criterion for determining whether a resident is currently living at a household location is commonly applied in this context to account for resident mobility.”.

On p7, section 2.2.3 the authors define Zi,j using a Bernoulli variable with probability p_j(x_i). Wouldn't it make sense to consider a hazard-based framework or derive the corresponding hazard function in terms of it's interpretation?

Thank you for your comment. Our study design consisted of two cross-sectional serological surveys conducted six months apart. We defined infections during this six-month period based on a comparison of antibody titres in paired serological samples from the two surveys. As we had no ongoing surveillance during this period (and because a significant proportion of infections are asymptomatic) we were only able to consider the outcome of whether there had been (at least one) infection event during the six-month period. Consequently, we did not have a time-to-event outcome and were therefore unable to consider a hazard-based framework.

Textual comments:– Please use \mbox for the correlation in section 2.2.1.

This has been corrected, thank you for pointing this out.

Reviewer #2 (Recommendations for the authors):Line 62 – typo? Analyze vs. analysing?

This has been corrected, thank you for pointing this typo out.

Figure 2 description – typo? … dh and dr are not mutually exclusive groups of explanatory [variables – missing?] and the same variables…

This has been corrected by adding in ‘variables’, thank you for pointing this typo out.

Line 277 – it would be nice to reference table 2 here so that the readers can see the full list of considered variables by group.

We have added the following text in the suggested sentence (Section 2.3.2): “(see Table 2 for the full list of considered variables by group)”

Regarding supplemental information, it would have been easier if the actual table or figure had been referenced vs. the file in which it could be found.

We have now added the Figure/Table reference for all references to supplemental information throughout the main text.

Reviewer #3 (Recommendations for the authors):It was a pleasure to review your manuscript.In my opinion the writing is excellent, the study design is clever and powerful (and must have been a lot of work!), and the spatial statistics are performed expertly.I do have a few suggestions, that I hope can either be easily refuted or can help to improve the analyses.Congratulations on this fantastic work.L65: I suggest writing DALYs in full, as not all readers will know what this is.

We have added the following text: “disability-adjusted life years (DALYs)”.

L94: I don't agree that there is an absence of methods (multilevel Bayesian models for example have been around for a while), but rather that they are rarely applied in this context.

We have changed ‘The absence of methods to formally integrate abundance into analyses of spillover mechanisms is an issue for rodent-borne zoonoses more widely’ to ‘the absence of methods applied to formally integrate to formally integrate abundance and spillover infection data is an issue for rodent-borne zoonoses more widely’.

L99: I find this particular unspecific use of abundance quite confusing, as this is already a very specific and well-defined ecological term. For example, what exactly is then meant by reservoir host abundance on L104? Is this the number of reservoir hosts, or the number of infected hosts, or the number of leptospires in the environment?If it is used as a measure of exposure to a disease of interest (L100), why not use a term like pathogen pressure, or just exposure? I strongly suggest using different words to describe actual abundance and pathogen-related abundance.

We agree that abundance in this context is well-defined as the density of reservoir hosts, however it is very commonly used to describe measures or estimates that are all proxies for ‘true abundance’. In our paper we wished to explicitly acknowledge the implications of using several imperfect abundance indices on our latent process. To make this clearer and more consistent we have changed the text to “We use the term ‘abundance’ here to denote all ecological processes that are associated with animal abundance and measured by abundance indices, for example animal presence, density and activity, and that may be useful to quantify exposure to a zoonotic disease of interest.”. We have also added “(as defined previously)” on line 106 to make it clear that we are consistently using this definition throughout the paper.

L115: The term rattiness is useful (and fun), but does it really represent leptospire pressure by rats if the model does not take into account leptospira prevalence/shedding in the rat populations? I agree that the presence/abundance of rats can be a decent proxy for the potential risk of leptospira spillover in locations with known presence of leptospira in the rat population, but I'm less inclined to accept that it is ok to define rattiness, which implies rat abundance, as a proxy for leptospiral contamination when the study did not measure the presence of leptospira in the rats.I see that this is mentioned in the discussion (L493). I think it might be more useful to add this information earlier on, at the place where the term rattiness is introduced.I agree that with such a high prevalence, it is reasonable to use rattiness as a proxy, but would still be wary: these 80% of rats are likely not distributed randomly across the area, as pathogen transmission is typically more spatially clustered. That means that 1 out of 5 local rat populations are not infected, which is definitely not negligible. That is an important caveat to highlight clearly, early on.

Thank you for this comment. We have added the following text on line 530 in the discussion to emphasise the possibility of spatial clustering of non-shedding rats:

“Despite this, non-shedding rats may be spatially clustered and future work would benefit from the collection of georeferenced rat infection data”. We have also edited the first sentence in on line 116 in the Introduction to the following: “The aim of this study was to develop a flexible modelling framework for zoonotic spillover to explore whether rattiness, acting as a proxy for local leptospiral contamination by Norway rats, can explain spatial heterogeneity in leptospiral transmission in a high-risk urban community in Brazil where 80% of rats are estimated to be actively shedding the bacteria.”

L208, 213: On L208, i is defined as a location, and on L213 as household location. I assume these represent the same location? If so, it might be best to be a bit more specific in the definition on L208, and add 'household', just so it's clear there is only one definition of a location.

We denote the full set of locations for which we have collected data (rat abundance sampling locations and participant household locations) with ‘i’, but index them as i = 1,…,N_r_ for rat index locations and i = N_r_+1,….,N_r_+N_h_ for household locations. This is important for our definition of R(x_i_) at all rat and human locations. We have tried to be consistent with this definition for the rat and human data – on line 219 we defined the human data to be at ‘a discrete set of locations X=…’. The reason for including the text ‘for individual j at household location i’ was to make it explicit that it was the household location which we were using for the human data.

L284: What is the rationale for choosing those specific knots?

We have added the following additional text in Appendix 2 to explain the use of the generalized additive model (GAM) plots to explore non-linear relationships between explanatory variables and human infection risk more explicitly:

“Single knot points were considered for age at 30 years (Figure S2A), education at 5 years (Figure S2C) and relative elevation at 20 metres (Figure S2D) based on the value of the explanatory variable at which the gradient of the relationship changed in these plots.”. This is already referenced in the main text “As before, non-linear relationships were modelled using linear splines based on the identified functional form. Age was modelled with a knot at 30 years old, education at 5 years and relative elevation at 20m (Figure S2 in Appendix 1)”.

Please see above for similar added text to clarify the same process for the rattiness explanatory variables.

L324: Kudos for citing the individual R packages (as one should, but often not done).

Thank you.